# Decision Tree Learning on Product Spaces

**Arshia Soltani Moakhar** [1]   **Faraz Ghahremani** [1]   **Kiarash Banihashem** [1]   **MohammadTaghi Hajiaghayi** [1]

## Abstract

Decision tree learning has long been a central topic in theoretical computer science, driven by its practical importance. A fundamental and widely used method for decision tree construction is the top-down greedy heuristic, which recursively splits on the most influential variable. Despite its empirical success, theoretical analysis of this heuristic has been limited. A recent breakthrough by Blanc et al. (ITCS, 2020) provided the first rigorous theoretical guarantees for the greedy approach, but only under the uniform distribution. We extend this analysis to the more general and practically relevant setting of arbitrary product distributions. Our main result shows that for any function $f$ computable by an optimal decision tree of size $s$, maximum depth $D_{\text{opt}}$, and average depth $\Delta_{\text{opt}}$, the greedy heuristic constructs an $\epsilon$-approximating tree whose size grows at most with $\exp\big(\Delta_{\text{opt}} D_{\text{opt}} \log(e/\epsilon)\big)$. In the special case where the optimal tree is a full binary tree, this bound improves upon the bound of Blanc et al. and holds under a strictly broader class of distributions. Moreover, we present an algorithm based on the top-down greedy heuristic that is entirely **parameter-free**—it requires no prior knowledge of the optimal tree's size or depth—offering a practical advantage over Blanc et al.'s method.

## 1. Introduction

Decision trees are a fundamental model in machine learning and algorithmic theory, valued for their interpretability, low evaluation cost, and broad empirical success. The core algorithmic task is to efficiently learn a decision tree approximating a target function. This challenge has driven a long line of theoretical work, yielding a rich collection of algorithms and complexity bounds across various settings (Ehrenfeucht & Haussler, 1989) (Information and Computation'89), (Saks & Wigderson, 1986) (FOCS'86), (Kushilevitz & Mansour, 1991) (STOC'91), (Mehta & Raghavan, 2002) (TCS'02), (O'Donnell & Servedio, 2007) (FOCS'07), (Gopalan et al., 2008) (STOC'08), (Blanc et al., 2020) (ITCS'20), (Brutzkus et al., 2020) (COLT'20), (Blanc et al., 2022a) (COLT'22), (Blanc et al., 2022b) (FOCS'22), (Moakhar et al., 2026) (ICLR'26), (Koch et al., 2023) (SODA'23), (Koch et al., 2024) (COLT'24).

Much of the theoretical analysis of decision tree algorithms has relied on the simplifying assumption that inputs are drawn from the uniform distribution over $\{0, 1\}^d$ (Mehta & Raghavan, 2002; Blanc et al., 2020; Gopalan et al., 2008; Blanc et al., 2022b). While analytically convenient, this assumption departs significantly from real-world settings, where data features often exhibit heterogeneous distributions. The class of general product distributions more accurately models such data, and understanding decision tree complexity in this setting has become a central line of inquiry (O'Donnell et al., 2005), (Kalai & Teng, 2008), (Brutzkus et al., 2020). Hence, we provide our analysis for this group of distributions.

In addition, most prior work introduces and analyzes algorithms for decision tree learning that have little to do with the algorithms used in practice. As a result, the strong performance of decision tree learning algorithms used in practice remains poorly understood. Recently, a paper by (Blanc et al., 2020) analyzes the top-down greedy approach to decision tree learning, which is closely related to the algorithms used in practice. We follow this work and analyze the same top-down greedy approach for a more general class of underlying data distributions, and derive performance guarantees that outperform the bounds of (Blanc et al., 2020).

### 1.1. The Greedy Heuristic and Influence-Based Splitting

One of the most natural and widely employed algorithmic approaches to building a decision tree is to proceed in a top-down, greedy fashion that operates by iteratively growing the tree. This method starts from a one-leaf decision tree and expands it by recursively selecting the "best" variable to query at one of the leaves.

This framework reduces the complex task of tree construction to a sequence of locally optimal decisions: choosing

---

[1]Department of Computer Science, University of Maryland, College Park, USA. Correspondence to: MohammadTaghi Hajiaghayi <hajiagha@umd.edu>.

*Proceedings of the 43$^{rd}$ International Conference on Machine Learning*, Seoul, South Korea. PMLR 306, 2026. Copyright 2026 by the author(s).

a splitting criterion at each node. Intuitively, a good variable to query should be one that is highly "relevant" to the function's output, thereby simplifying the remaining sub-problems as much as possible. A canonical measure of such relevance is the influence of a variable.

Let $\mu = \mu_1 \times \cdots \times \mu_n$ be a product distribution over $\{0,1\}^n$, where for each coordinate $i \in [n]$, a bit is chosen to be 1 with probability $p_i$ and 0 with probability $1 - p_i$. The influence of the $i$-th variable on a function $f : \{0,1\}^n \to \{\pm 1\}$ with respect to $\mu$ is defined as:

$$\text{Inf}_i^\mu(f) := \Pr_{x \sim \mu}[f(x) \neq f(x^{(i)})]$$

where $x^{(i)}$ denotes $x$ with its $i$-th coordinate re-randomized according to $\mu_i$.

The top-down heuristic we analyze, formalized as *Build-TopDownDT* by (Blanc et al., 2020), operates by iteratively growing a decision tree. In each step, the algorithm scores every leaf in the current tree. The score for a leaf $l$ is defined as the probability of an input reaching that leaf, multiplied by the influence of the *most* influential variable for the sub-function at that leaf. The algorithm then selects the leaf with the highest score and splits it by adding a new decision node corresponding to its most influential variable. This process is repeated until there exists a label assignment to the tree's leaves that is an $\epsilon$-approximation of the target function. This influence-based splitting criterion is the basis of celebrated algorithms like ID3 (Quinlan, 1986), C4.5 (Quinlan, 2014), and CART (Breiman et al., 2017), which use impurity measures such as entropy or the Gini index that are closely related to influence.

Blanc et al. (Blanc et al., 2020) show that for any function $f$ with an optimal decision tree of size $s$, the greedy heuristic produces an $\epsilon$-approximating tree of size at most $s^{O(\log(s/\epsilon) \log(1/\epsilon))}$. However, their analysis is intrinsically tied to the rich Fourier-analytic structure of the Boolean hypercube under the uniform distribution. Key steps in their proof leverage powerful inequalities relating variance and influence that are well-established in the uniform distribution setting. This reliance on uniform distribution leaves a critical gap between theory and practice, as real-world data is rarely uniformly distributed. We fill this gap by generalizing the analysis to arbitrary product distributions.

## 1.2. Our Contributions

Our paper bridges the gap between the theoretical analysis of decision tree learning and its practical application by extending the analysis of the top-down greedy heuristic to the general product space setting. Our main result demonstrates that the performance guarantees of the influence-based greedy heuristic are robust and hold even without the structural convenience of the uniform distribution, a setting where prior rigorous guarantees were largely limited.

Our analysis focuses on a top-down greedy heuristic, an approach that closely mirrors the simple and practical algorithms used widely in machine learning. This contrasts with prior theoretical works that often rely on more complex mechanisms. Because these approaches diverge from the heuristics used in practice, they offer limited insight into the reasons for the strong empirical performance of common algorithms. For instance, some foundational algorithms build the tree bottom-up or involve brute-force components over potential splits (Blanc et al., 2022b; Ehrenfeucht & Haussler, 1989), while other recent advances achieve faster runtimes by designing more intricate algorithms that deviate from the simple greedy heuristic. Our contribution is to provide rigorous performance guarantees for the classic greedy method itself, helping to bridge this gap between theoretical understanding and practical application.

Our main theoretical result is stated below:

**Theorem 1.1** (Main Result)**.** *Let $f : \{0,1\}^n \to \{\pm 1\}$ be any Boolean function that can be computed by a decision tree $T_{opt}$ of depth $D_{opt}$ and average depth $\Delta_{opt}$ under an arbitrary product distribution $\mu$. For any $\epsilon > 0$, Algorithm 1 returns a decision tree that $\epsilon$-approximates $f$ and has size at most:*

$$\max\left( \left( \frac{e \cdot \Delta_{opt}}{\epsilon D_{opt}} \right)^{\Delta_{opt} D_{opt}}, e^{\Delta_{opt} D_{opt}} \right)$$

We note that in the specific case of a full binary tree, our performance bound simplifies to $s^{\log s \cdot \log(e/\epsilon)}$, or more generally in balanced decision trees when $\Delta_{opt}$ and $D_{opt}$ are in $O(\log s)$ we get $s^{O(\log s \cdot \log(e/\epsilon))}$. This bound is tighter and improves upon the one established by (Blanc et al., 2020), but also applies to a broader class of distributions.

The two parameters $D_{opt}$ and $\Delta_{opt}$ enter Theorem 1.1 for distinct reasons. The maximum depth $D_{opt}$ enters when relating total influence to variance (Lemma 4.2). The average depth $\Delta_{opt}$ enters through the max-influence inequality $\max_i \text{Inf}_i(f) \geq \text{Var}(f)/\Delta(T)$ of O'Donnell et al. (2005). Two regimes are worth highlighting. In the *balanced* regime ($D_{opt} = \Delta_{opt} = \log s$), the bound becomes $s^{\log(s) \log(e/\epsilon)}$, which is quasi-polynomial in $s$. In the *path-like* regime, $D_{opt}$ can be as large as $n$ while the average depth $\Delta_{opt}$ remains a small constant (see the path-tree example in Appendix E.2); the product $\Delta_{opt} D_{opt}$ is then exponentially smaller than $D_{opt}^2$, so the mixed dependence on the two depth parameters is strictly tighter than a bound stated in $D_{opt}$ alone.

In addition, we provided the first practical parameter-free implementation of *BuildTopDownDT*. In contrast to (Blanc et al., 2020), we do not assume any prior knowledge of the optimal tree's size ($s$) or depth, which is a notable practical advantage.

### 1.3. Broader Context of Decision Tree Learning

Our work is situated within a broader line of research on learning decision trees in quasi-polynomial time. The seminal work of Ehrenfeucht and Haussler (Ehrenfeucht & Haussler, 1989) provided a distribution-free algorithm for properly learning a decision tree of size $s$ in time $n^{O(\log s)}$. Subsequent work has largely focused on achieving faster runtime, typically by restricting the problem to the uniform distribution. Mehta et al. (Mehta & Raghavan, 2002) introduced a dynamic programming approach with a runtime of $n^{O(\log(s/\epsilon))}$. More recently, Blanc et al. (Blanc et al., 2022b) achieved a nearly-polynomial runtime of $n^{O(\log \log n)}$ by designing a more complex algorithm that deviates from the simple greedy heuristic, instead considering a polylogarithmic number of influential variables at each step.

While these works argue that they can be used to understand the practical success of decision trees, they only analyzed brute-force algorithms which deviate significantly from practical algorithms. In contrast to these works that design novel or more complex algorithms for the uniform case, our contribution is to provide the first rigorous performance guarantees for the classic, simple greedy heuristic in the more general setting of arbitrary product distributions. This analysis addresses a long-standing open question and helps bridge the gap between the theoretical understanding of fundamental learning algorithms and their widespread practical use.

## 2. Related Works

(Ehrenfeucht & Haussler, 1989) presented an algorithm for learning decision trees of size $s$ in time $n^{O(\log s)}$, which is robust to arbitrary data distributions. The algorithm employs a backtracking approach that exhaustively searches over all decision trees of small rank.

The landscape of quasi-polynomial time algorithms was further developed by (Mehta & Raghavan, 2002), who presented a dynamic programming approach for learning decision trees specifically under the uniform distribution. Their algorithm constructs an optimal tree subject to size and height constraints by exhaustively building solutions for all partial assignments up to a given depth. This bottom-up, exhaustive search guarantees a hypothesis of size at most $s$ but has a runtime of $n^{O(\log(s)/\epsilon)}$. In contrast, our work analyzes a top-down, greedy heuristic that can be generalized beyond the uniform distribution.

In a related line of work, (Blanc et al., 2022b) focused on improving the runtime for learning decision trees under the uniform distribution. They proposed a more sophisticated algorithm that achieves an almost-polynomial runtime of $(s/\epsilon)^{O(\log \log(s/\epsilon))}$. Their approach departs from the standard greedy heuristic by evaluating a broader set of $polylog(s)$ influential variables at each step. This is made possible by a structural result showing that any decision tree can be pruned such that every remaining node has sufficiently high influence.

The work of (Blanc et al., 2022b) and our own address complementary but orthogonal questions. While their focus is on designing a faster, more intricate algorithm tailored to the uniform distribution, our work analyzes the performance of the original, simpler greedy heuristic in the more general—and practically relevant—setting of arbitrary product distributions.

**Classical greedy implementations.** Greedy top-down decision tree induction has a long history in practice through algorithms such as ID3 (Quinlan, 1986), C4.5 (Quinlan, 2014), and CART (Breiman et al., 2017). These methods all share the same skeleton as Algorithm 1: starting from a single leaf, they recursively grow the tree by splitting on a locally-best feature and stopping at a leaf when an impurity-based criterion (entropy gain for ID3/C4.5, Gini impurity for CART) drops below a threshold. The impurity measures used by these algorithms are closely related to coordinate influence under product distributions: entropy and Gini both quantify the variance reduction achieved by a split, and our analysis shows that influence-based splitting — a direct theoretical analogue — enjoys provable guarantees in the same regime. Most prior theoretical analyses, however, study non-greedy variants and therefore do not directly speak to the success of these classical algorithms; our results help fill that gap.

**Lower bounds and hardness.** Two recent results clarify how much improvement on Algorithm 1 is possible in principle. Koch et al. (2023) prove superpolynomial lower bounds for decision tree learning and testing under the uniform distribution: there is no $\text{poly}(s, n, 1/\epsilon)$-time algorithm that produces a decision tree of size $\text{poly}(s)$ approximating the target. This rules out a fully polynomial-size guarantee even with membership queries, so the quasi-polynomial dependence on $s$ in our bound is, up to constants in the exponent, near-tight. The follow-up work of Koch et al. (2024) strengthens this to a *superconstant inapproximability* result: it is NP-hard to find a tree of size at most $s^c$ for any constant $c$, even with membership queries to the target.

## 3. Preliminaries

In this section, we first describe our learning setting and then define the main concepts and notations used throughout the paper.

**Setting.** We assume query access to the target function $f : \{0,1\}^n \to \{\pm 1\}$. In particular, we allow the following

operations:

- *Label Query:* For any input $x \in \{0,1\}^n$, query $f(x)$.

- *Random Sample:* Draw a sample $x \sim \mu$, where $\mu$ is the underlying product distribution over $\{0,1\}^n$.

**Notation.** We begin by introducing the key concepts and quantities that will be used throughout our analysis. Our algorithm operates over a structure we refer to as a *bare tree* $T^\circ$, which is a decision tree where each internal node queries a single variable, but the leaves remain unlabeled. Once the structure of $T^\circ$ is fixed, we derive its *$f$-completion* by assigning to each leaf the majority label of the target function $f$, restricted to the leaf's region under distribution $\mu$.

For any node $v$ in the tree, we define the *subfunction* $f_v$ as the restriction of $f$ to the region associated with $v$. The *error* of a subfunction, denoted $\mathrm{error}(f_v, \pm 1)$, measures the minimum probability of misclassification:

$$\mathrm{error}(f_v, \pm 1) = \min \left( \Pr[f_v(x) = -1], \Pr[f_v(x) = +1] \right).$$

The *size* of a tree $T$, denoted $\mathrm{size}(T)$, is its total number of leaves; equivalently, $\mathrm{size}(T)$ equals the number of internal nodes of $T$ plus one. To analyze the structure of the tree, we use both its *depth*—the maximal path length from root to any leaf, denoted $d(T)$—and its *average depth*, defined as:

$$\Delta(T) = \sum_{v \in \mathrm{leaves}(T)} p_v \cdot \mathrm{depth}(v).$$

where $p_v$ is defined as $\Pr_{x \sim \mu}[x \text{ reaches } v]$, the probability of a random sample reaching $v$. Note that by the above definition for $\Delta(T)$, we also have $\Delta(T) = \sum_{v \in T} p_v$. At each leaf $v$, we define its *score* as the product of the probability mass reaching it and the most influential coordinate:

$$\mathrm{Score}(v) = p_v \cdot \max_i \mathrm{Inf}_i^\mu(f_v).$$

We define the *total influence* of the function $f$ itself as the sum of influence across all dimensions, meaning $\mathrm{Inf}(f) = \sum_{i=1}^n \mathrm{Inf}_i(f)$. Finally, the *cost* of a bare tree $T^\circ$ is the sum of total influence over all leaves:

$$\mathrm{cost}(T^\circ) = \sum_{v \in \mathrm{leaves}(T^\circ)} p_v \cdot \mathrm{Inf}(f_v).$$

# 4. Main Theoretical Results

In this section, we present our main theoretical contributions. We begin by formally defining the top-down, greedy algorithm we analyze. We then state our primary theorem, which provides a performance guarantee on the size of the

decision tree this algorithm constructs for any function on a product space. The remainder of the section is dedicated to the proof of this theorem, which we build through a series of key lemmas that establish the relationships between a tree's approximation error, a potential function we call "cost", and the guaranteed progress the algorithm makes at each iteration.

## 4.1. The Top-Down Greedy Algorithm

The algorithm we analyze, "BuildTopDownDT", is a formalization of the widely used greedy heuristic for decision tree induction. It begins with a trivial tree (a single leaf) and iteratively expands it. At each step, it identifies the "best" leaf to split by calculating a score for each leaf. This score is the product of the probability of an input reaching the leaf and the influence of the most influential variable for the subfunction at that leaf. The leaf with the highest score is then replaced by a decision node that queries its most influential variable. This process continues until there exists a labeling of the tree leaves that is an $\epsilon$-approximation of the target function $f$. We call the optimal leaf labeling $f$-completion of $T^\circ$. The procedure is detailed in Algorithm 1, from (Blanc et al., 2020).

---

**Algorithm 1** Top-Down Heuristic for Decision Tree Induction, from (Blanc et al., 2020)

---

1: **function** BuildTopDownDT($f, \epsilon$)
2: Initialize $T^\circ$ to be a single-leaf tree.
3: **while** $f$-completion of $T^\circ$ is not an $\epsilon$-approximation of $f$ **do**
4:     *// Score all leaves*
5:     **for** each leaf $l$ in $T^\circ$ **do**
6:         Let $x_{i(l)}$ be the most influential variable of the subfunction $f_l$.
7:         $\mathrm{Score}(l) \leftarrow \Pr_{x \sim \mu}[x \text{ reaches } l] \cdot \mathrm{Inf}_{i(l)}(f_l)$
8:     **end for**
9:     *// Split the best leaf*
10:    Let $l^*$ be the leaf with the highest score.
11:    Grow $T^\circ$ by replacing leaf $l^*$ with a query to $x_{i(l^*)}$.
12: **end while**
13: **return** The $f$-completion of $T^\circ$.

---

## 4.2. Performance Guarantee for Product Distributions

Our main result in Theorem 1.1 demonstrates that this natural greedy heuristic produces a decision tree of quasi-polynomial size in the balanced regime, generalizing the prior analysis from the uniform distribution to any product distribution. We note that some quasi-polynomial dependence on the optimal tree size $s$ is unavoidable in general: the superpolynomial lower bounds of Koch et al. (2023) rule out a fully polynomial-size guarantee for decision tree learning, even with membership queries.

Moreover, for full binary trees of size $s$, we have $D_{opt} = \Delta_{opt} = \log s$; therefore, for full decision trees this bound is $\left(\frac{e}{\epsilon}\right)^{\log^2 s} = \exp\left(\log^2(s)\log(1/\epsilon)\right) = s^{\log(s)\log(1/\epsilon)}$. This bound is slightly better than (Blanc et al., 2020) and holds for any product-space input distribution, generalizing the uniform distribution.

### 4.3. Proof of the Main Theorem

To prove Theorem 1.1, our strategy is to analyze the behavior of a potential function, the cost of the bare tree $T^\circ$, as the algorithm runs. We show that this cost is an upper bound on the tree's error (Lemma 4.1) and that it starts at a bounded value (Lemma 4.2). We then prove that at each step of the algorithm, the cost decreases by the influence of the selected leaf (Lemma 4.3) which is a significant amount (Lemma 4.4, and Lemma 4.5). By bounding the number of steps required for the cost to fall below $\epsilon$, we can bound the final size of the tree (Lemma 4.6, and Lemma 4.7).

The first lemma establishes that the cost of a tree is an upper bound on its approximation error. This ensures that if we drive the cost down to $\epsilon$, the algorithm terminates.

**Lemma 4.1** (Error is Bounded by Cost). *For any bare tree $T^\circ$ and function $f$, the error of its $f$-completion is bounded by its cost:*

$$\text{error}(T^\circ, f) \leq \text{cost}(T^\circ)$$

*Proof sketch of Lemma 4.1, full proof is in Appendix C.2.*
It is enough to prove the bound for each leaf subfunction $g$, namely

$$\text{error}(g, \pm 1) \leq \text{Inf}(g),$$

and then sum over leaves.

We show this inequality by induction on the number of variables of $g$. For $n = 1$ it is a direct check: the error is the minority mass, while $\text{Inf}(g)$ is exactly the probability that flipping the lone bit changes the value. For $n > 1$, fix one input bit (say the last bit). Let $g_0$ be the function you get by setting that bit to 0, and let $g_1$ be the function you get by setting that bit to 1. By the inductive hypothesis, $\text{error}(g_b, \pm 1) \leq \text{Inf}(g_b)$ for $b \in \{0, 1\}$. Using the standard decomposition of total influence under restriction and the corresponding decomposition of the error of $g$ in terms of the errors of $g_0, g_1$ and their disagreement, a short algebraic rearrangement yields $\text{error}(g, \pm 1) \leq \text{Inf}(g)$. Summing this bound over leaves gives $\text{error}(T^\circ, f) \leq \text{cost}(T^\circ)$. $\quad\square$

The next lemma provides an upper bound for the initial value of the cost function.

**Lemma 4.2** (Total Influence vs. Variance and Depth). *For any function $f : \{0, 1\}^n \to \{\pm 1\}$ represented by a decision tree $T$ with depth $D(T)$, we have:*

$$\text{Inf}(f) \leq D(T) \cdot \text{Var}(f) \quad \text{and} \quad \text{Inf}(f) \leq \Delta(T)$$

The proof of the previous lemma is in Appendix C.3. The next lemma is the engine of our analysis, showing that each split reduces the cost by precisely the Score of the leaf that was split.

**Lemma 4.3** (Cost Reduction per Step). *Let $T^\circ$ be a bare tree, and let $T'^\circ$ be the tree obtained by splitting a leaf $v \in \text{leaves}(T^\circ)$ on its most influential variable, $i(v)$. The cost of the new tree is:*

$$\text{cost}(T'^\circ) = \text{cost}(T^\circ) - \text{Score}(v)$$

*Proof sketch of Lemma 4.3, full proof is in Appendix C.4.*
The cost of a tree is the sum of costs contributed by its leaves. When we split leaf $v$ into new leaves $v_0$ and $v_1$, the leaves in $\text{leaves}(T^\circ) \setminus \{v\}$ are unaffected. We then show that when a node splits into two nodes by splitting on variable $i$, the sum of total influence of new nodes is equal to the old node's total influence minus $\text{Inf}_i(f_v)$. $\quad\square$

We now provide two different lower bounds on the score, corresponding to the two phases of our analysis.

**Lemma 4.4** (Score Lower Bound When Error is High). *Let $T^\circ$ be the bare tree after $j - 1$ steps of Algorithm 1. If the algorithm has not terminated, then the leaf $l^*$ chosen by the algorithm has a score of at least:*

$$\text{Score}(l^*) \geq \frac{2\epsilon}{j \cdot \Delta_{opt}}$$

*Proof of Lemma 4.4.* Since the total error is greater than $\epsilon$ and there are $j$ leaves, by an averaging argument, there must exist at least one leaf $l$ where its contribution to the error is at least $\epsilon/j$. That is, $p_l \cdot \text{error}(f_l, \pm 1) \geq \frac{\epsilon}{j}$. Since $\text{Var}(f_l) \geq 2 \cdot \text{error}(f_l, \pm 1)$ (Lemma C.2), we have $p_l \cdot \text{Var}(f_l) \geq \frac{2\epsilon}{j}$. By Lemma C.1 applied to $f_l$, we have

$$\text{Score}(l) = p_l \cdot \max_i \text{Inf}_i(f_l) \geq p_l \cdot \frac{\text{Var}(f_l)}{\Delta(f_l)}.$$

Since the algorithm chooses the leaf $l^*$ with the maximum score, we have $\text{Score}(l^*) \geq \text{Score}(l)$. Therefore,

$$\text{Score}(l^*) \geq \text{Score}(l) \geq p_l \text{Var}(f_l) \cdot \frac{1}{\Delta(f_l)}$$

$$\geq \frac{2\epsilon}{j \cdot \Delta_{opt}}.$$

$\quad\square$

**Lemma 4.5** (Score Lower Bound When Cost is High). *Let $T^\circ$ be a bare tree with $j$ leaves. The leaf $l^*$ chosen by the Algorithm 1 has a score of at least:*

$$\text{Score}(l^*) \geq \frac{\text{cost}(T^\circ)}{j \cdot D_{opt}\Delta_{opt}}$$

*Proof of Lemma 4.5.* The algorithm chooses $l^*$ to maximize the score. The maximum is at least the average score.

$$\text{Score}(l^*) \geq \frac{1}{j} \sum_{l \in \text{leaves}(T^\circ)} \text{Score}(l) = \frac{1}{j} \sum_l p_l \cdot \max_i \text{Inf}_i(f_l)$$

By Lemma C.1, we have $\max_i \text{Inf}_i(f_l) \geq \text{Var}(f_l)/\Delta(f_l)$. Moreover, Lemma 4.2 gives $D(f_l)\text{Var}(f_l) \geq \text{Inf}(f_l)$, and hence $\text{Var}(f_l) \geq \text{Inf}(f_l)/D(f_l)$.

$$\text{Score}(l^*) \geq \frac{1}{j} \sum_l p_l \cdot \frac{\text{Var}(f_l)}{\Delta(f_l)} \geq \frac{1}{j} \sum_l p_l \cdot \frac{\text{Inf}(f_l)}{D(f_l)\Delta(f_l)}$$

Since $D(f_l) \leq D_{opt}$ and $\Delta(f_l) \leq \Delta_{opt}$ for any $f_l$:

$$\text{Score}(l^*) \geq \frac{1}{jD_{opt}\Delta_{opt}} \sum_l p_l\text{Inf}(f_l) = \frac{\text{cost}(T^\circ)}{jD_{opt}\Delta_{opt}}$$

$\square$

We now combine these lemmas to prove Theorem 1.1. We analyze the algorithm's progress in two phases: first, when the cost is high (Lemma 4.6), and second, when the cost is low but the error remains above $\epsilon$ (Lemma 4.7) with proofs provided in Appendix C.5, and Appendix C.6 respectively.

**Lemma 4.6** (Phase 1: High Cost Reduction). *Let $J_1$ be the number of steps required for the cost $C_j$ to drop from its initial value to at most $\epsilon D_{opt}$ in Algorithm 1. Then:*

$$J_1 \leq \max\left(\left(\frac{\Delta_{opt}}{\epsilon D_{opt}}\right)^{D_{opt}\Delta_{opt}}, 1\right)$$

**Lemma 4.7** (Phase 2: Termination). *Let $J_1$ be the step at which the cost drops to $\epsilon D_{opt}$. The Algorithm 1 guarantees termination at a total number of steps $J_{total}$ satisfying:*

$$J_{total} \leq J_1 \cdot e^{\Delta_{opt}D_{opt}}$$

*Proof of Theorem 1.1.* The size of the final tree is $J_{total}+1$. By Lemma 4.7,

$$J_{total} \leq J_1 \cdot e^{\Delta_{opt}D_{opt}}.$$

Lemma 4.6 gives

$$J_1 \leq \max\left(\left(\frac{\Delta_{opt}}{\epsilon D_{opt}}\right)^{\Delta_{opt}D_{opt}}, 1\right),$$

so

$$J_{total} \leq \max\left(\left(\frac{e \cdot \Delta_{opt}}{\epsilon D_{opt}}\right)^{\Delta_{opt}D_{opt}}, e^{\Delta_{opt}D_{opt}}\right).$$

$\square$

# 5. Algorithmic Implications and Practical Implementation

Algorithm 1 and the analysis presented so far assume exact computation of probabilities and influences. In practice, these quantities must be estimated from samples. This section addresses the algorithmic consequences of this reality. First, we show that the performance guarantee is robust to estimations, requiring only that we select a "good enough" leaf to split, not necessarily the absolute best. Second, we detail how to estimate the scores and the tree's error to guide the algorithm and its termination condition, providing sample complexity bounds for each step.

### 5.1. A Robust Algorithm with Approximate Scores

We first show that the algorithm's performance guarantee gracefully degrades if we only find a leaf whose score is a constant fraction of the maximum score.

**Theorem 5.1.** *Let the conditions of Theorem 1.1 hold. If at each step the algorithm selects a leaf $l'$ to split such that $\text{Score}(l') \geq \frac{1}{4}\max_l \text{Score}(l)$, then it returns an $\epsilon$-approximating tree of size at most:*

$$\max\left(\left(\frac{e \cdot \Delta_{opt}}{\epsilon D_{opt}}\right)^{4\Delta_{opt}D_{opt}}, e^{4\Delta_{opt}D_{opt}}\right)$$

*Proof Sketch of Theorem 5.1, full proof is in Appendix C.7.* In contrast to Theorem 1.1, this time our lower bound on the maximum score is $1/4$ of the lower bounds used to prove that theorem. In particular when decreasing cost from $\Delta$ to $\epsilon D_{opt}$ instead of our previous bound $\frac{\text{cost}}{(j+1)\Delta D_{opt}}$ in Lemma 4.5 we get $\frac{\text{cost}}{(j+1)(4\Delta)D_{opt}}$. Similarly when decreasing cost further to $\epsilon$ instead of $\frac{\epsilon}{(j+1)\Delta}$ lower bound in Lemma 4.4 we get $\frac{\epsilon}{(j+1)(4\Delta)}$. In both of these cases it is algebraically equivalent to having a 4 times larger $\Delta$ which means the maximum number of steps will be

$$\left(\frac{e \cdot \text{Inf}(f)}{\epsilon D_{opt}}\right)^{4\Delta D_{opt}} \leq \left(\frac{e \cdot \Delta}{\epsilon D_{opt}}\right)^{4\Delta D_{opt}}.$$

$\square$

### 5.2. Estimating Leaf Scores

Theorem 5.1 is key to the practical applicability of our algorithm, as it guarantees correctness even when leaf scores are estimated with some imprecision. In this section, we describe how to estimate these scores from samples with sufficient accuracy.

To execute Algorithm 1, we must estimate the score $\text{Score}(l, i) = p_l \cdot \text{Inf}_i(f_l)$, since both $p_l$ and the influence $\text{Inf}_i(f_l)$ are unknown. Fortunately, as shown in Theorem 5.1, it suffices to select a leaf-variable pair whose score is within

a constant factor (e.g., $1/4$) of the maximum. We observe that the score can be rewritten as:

$$\text{Score}(l, i) = \Pr_{x \sim \mu} [x \text{ reaches } l \text{ and } f(x) \neq f(x^{(i)})],$$

where $x^{(i)}$ is defined by re-randomizing the $i$-th coordinate of $x$ independently according to the marginal distribution $\mu_i$. To estimate this probability, we sample $x \sim \mu$, and for each coordinate $i \in [n]$, independently sample $x' \sim \mu$ and define $x^{(i)}$ by setting $x_j^{(i)} = x_j$ for $j \neq i$ and $x_i^{(i)} = x_i'$. This procedure correctly simulates a re-randomization of the $i$-th coordinate. Based on a multiset $E_i$ of such sampled pairs $(x, x^{(i)})$, we define the empirical estimator:

$$\widehat{\text{Score}}(l, i, E_i) =$$

$$\frac{1}{|E_i|} \sum_{(x, x^{(i)}) \in E_i} \mathbf{1}[x, x^{(i)} \text{ reach } l] \cdot \mathbf{1}[f(x) \neq f(x^{(i)})]. \quad (1)$$

We formalize its unbiasedness below and provide the proof in Appendix C.8.

**Lemma 5.2** (Unbiasedness of the Score Estimator). *The estimator $\widehat{\text{Score}}(l, i, E_i)$ is an unbiased estimator of the true score $\text{Score}(l, i)$.*

To ensure the algorithm selects a leaf-variable pair with score at least $1/4$ of the optimal, it suffices to ensure that all empirical scores are accurate within a factor of 2. We achieve this using multiplicative Chernoff bounds. Since each term in the estimator is a bounded random variable in $\{0, 1\}$, standard concentration results apply.

Let $t$ be a score threshold. Then for any pair $(l, i)$, we have:

$$\begin{cases} \text{If } \text{Score}(l, i) \geq t, & \Pr\left[\widehat{\text{Score}}(l, i, E_i) \leq \frac{t}{2}\right] \leq \exp\left(-\frac{|E_i| t}{8}\right) \\ \text{If } \text{Score}(l, i) \leq \frac{t}{4}, & \Pr\left[\widehat{\text{Score}}(l, i, E_i) \geq \frac{t}{2}\right] \leq \exp\left(-\frac{|E_i| t}{12}\right) \end{cases}$$

To ensure correct selection, we must guarantee that all leaf-variable pairs with score above $t$ are not underestimated (i.e., fall below $t/2$), and those with score below $t/4$ are not overestimated (i.e., exceed $t/2$). Note that in step $j$ the tree has $j + 1$ leaves so the number of leaf-variable pairs is $(j + 1)n$. This gives us:

$$\Pr[\text{any error at step } j] \leq (j + 1)n \cdot \exp(-\min_i |E_i| \cdot t/12).$$

Applying Lemma 4.4, which guarantees $t \geq \frac{\epsilon}{(j+1)\Delta}$, we obtain the lower bound:

$$\Pr[\text{failure at step } j] \leq (j + 1)n \cdot \exp(-\frac{m_j \cdot \epsilon}{12(j + 1)n}),$$

where $m_j := \min_i |E_i|$ is the minimum number of samples per variable at step $j$. Note that $\Delta(T) \leq D(T) \leq n$.

To ensure an overall failure probability of at most $\delta/2$ across all steps, it suffices to ensure that the failure probability at step $j$ is at most $\frac{\delta}{4j^2}$, since $\sum_{j=1}^{\infty} \frac{\delta}{4j^2} \leq \frac{\delta}{2}$. This leads to the sample complexity requirement:

$$m_j \geq M_S(j, \delta, \epsilon, n) := \frac{12(j + 1)n}{\epsilon} \cdot \ln\left(\frac{4j^2(j + 1)n}{\delta}\right).$$

We define the function $M_S(j, \delta, \epsilon, n)$ for convenience. At each step $j$, we ensure that each multiset $E_i$ contains at least $M_S(j, \delta, \epsilon, n)$ samples. If not, we augment $E_i$ by drawing additional random samples as described above.

### 5.3. Estimating Error

To estimate the error of our bare decision tree $T^\circ$, we first assign labels to its leaves to create a labeled tree $T$, and then we estimate the error of $T$. This process involves two steps, each requiring a sufficiently large sample set.

PART 1: ASSIGNING LABELS

In step $j$ of our algorithm, we use a set of $\hat{m}_j$ i.i.d. samples to label the leaves of $T^\circ$. For each leaf, we assign the label held by the majority of samples that fall into it. This ERM (Empirical Risk Minimization) procedure results in the labeled tree $T$. The following lemma bounds the excess error between our tree $T$ and the optimal labeling $T^*$.

**Lemma 5.3.** *Let $T^\circ$ be a decision tree with $l$ leaves. Let $T$ be the tree obtained by assigning a label from $\{-1, +1\}$ to each leaf of $T^\circ$ based on the majority vote of $\hat{m}_j$ samples drawn i.i.d. from a distribution $D$ and labeled by a function $f$. Let $T^*$ be the optimal labeling of $T^\circ$. For any $\delta \in (0, 1)$, with probability at least $1 - \frac{\delta}{8j^2}$, the following bound holds:*

$$|\text{error}(T, f) - \text{error}(T^*, f)| \leq \sqrt{\frac{2(l \ln(2) + \ln(16j^2/\delta))}{\hat{m}_j}}$$

*Proof sketch of Theorem 5.3, full proof is in Appendix C.9.* The problem of assigning labels to the $l$ leaves of the bare tree $T^\circ$ can be framed within the statistical learning framework. The set of all possible labeled trees constitutes our hypothesis space, denoted by $\mathcal{H}$, of size $|\mathcal{H}| = 2^l$. The majority vote procedure is an application of Empirical Risk Minimization (ERM) over this space. The result follows from a standard uniform convergence bound for finite hypothesis spaces, applied to the ERM hypothesis. The bound is typically two-sided, so we state it with an absolute value. $\square$

PART 2: ESTIMATING THE ERROR OF THE LABELED TREE

After creating the labeled tree $T$, we must estimate its true error, $\text{error}(T, f)$, to decide if it is a "good enough" hypothesis. To do this, we draw a second, independent set of $m_j''$ samples, denoted $S$, and compute the empirical error $\widehat{\text{error}}_S(T)$. The following lemma shows that if the sample sizes are large enough, this empirical estimate is a reliable indicator for termination.

**Lemma 5.4** (Reliable Error Estimation). *Let $\epsilon > 0$ be a target accuracy. Let $T$ be the tree from Part 1, constructed with $\hat{m}_j$ samples such that $\sqrt{\frac{2(l\ln(2)+\ln(16j^2/\delta))}{\hat{m}_j}} \leq \frac{\epsilon}{8}$. Let $S$ be a new set of $m_j''$ i.i.d. samples, where $m_j'' \geq \frac{32}{\epsilon^2}\ln(\frac{16j^2}{\delta})$. Let the event in Lemma 5.3 hold. Then:*

$$
\begin{cases}
\text{If } \text{error}(T^*, f) \leq \frac{\epsilon}{2} & \Pr\left[\widehat{\text{error}}_S(T) > \frac{3\epsilon}{4}\right] \leq \frac{\delta}{8j^2}. \\
\text{If } \text{error}(T, f) > \epsilon & \Pr\left[\widehat{\text{error}}_S(T) \leq \frac{3\epsilon}{4}\right] \leq \frac{\delta}{8j^2}.
\end{cases}
$$

*Proof sketch of Lemma 5.4, full proof is in Appendix C.10.* Assuming the event in Lemma 5.3 holds, the condition on $\hat{m}_j$ implies $|\text{error}(T, f) - \text{error}(T^*, f)| \leq \epsilon/8$.

If $\text{error}(T^*, f) \leq \epsilon/2$, then $\text{error}(T, f) \leq 5\epsilon/8$, and Hoeffding's inequality gives $\Pr[\widehat{\text{error}}_S(T) > 3\epsilon/4] \leq \exp(-m_j''\epsilon^2/32) \leq \delta/(8j^2)$ under the stated lower bound on $m_j''$. If instead $\text{error}(T, f) > \epsilon$, then Hoeffding's inequality similarly yields $\Pr[\widehat{\text{error}}_S(T) \leq 3\epsilon/4] \leq \exp(-m_j''\epsilon^2/8) \leq \delta/(8j^2)$. $\square$

OVERALL GUARANTEES

By combining the results of Lemma 5.3 and Lemma 5.4, we can state strong guarantees on the behavior of our algorithm over its entire execution. The algorithm terminates at the first step $j$ for which $\widehat{\text{error}}_S(T) \leq \frac{3\epsilon}{4}$.

First, we apply a union bound over all possible steps $j = 1, 2, \ldots$. The probability that the high-probability event in Lemma 5.3 or Lemma 5.4 fails at any step $j$ is at most $\sum_{j=1}^{\infty}(\frac{\delta}{8j^2} + \frac{\delta}{8j^2}) = \frac{\delta}{4}\sum_{j=1}^{\infty}\frac{1}{j^2} \leq \frac{\delta}{2}$. Thus, with probability at least $1 - \frac{\delta}{2}$, the conclusions of both lemmas hold for all steps $j$ simultaneously if for $\hat{m}_j$ we have $\sqrt{\frac{2(l\ln(2)+\ln(16j^2/\delta))}{\hat{m}_j}} \leq \frac{\epsilon}{8}$ and $m_j'' \geq \frac{32}{\epsilon^2}\ln(\frac{16j^2}{\delta})$.

We define $M_{EE}(j, \epsilon, \delta)$ as this value. We similarly define the bound for $\hat{m}_j$ as $M_{LL}(j, \epsilon, \delta) := \frac{128((j+1)\ln(2)+\ln(16j^2/\delta))}{\epsilon^2}$. Assuming we are in this high-probability event, we have the following guarantees:

1. **Correctness:** The algorithm will not terminate with a bad tree with high probability. If the algorithm terminates at step $j$, it must be that $\widehat{\text{error}}_S(T) \leq \frac{3\epsilon}{4}$. By Lemma 5.4 (Case 2), this is highly unlikely to happen if $\text{error}(T, f) > \epsilon$. Therefore, upon termination, we can be confident that $\text{error}(T, f) \leq \epsilon$.

2. **Termination:** The algorithm is guaranteed to terminate if the bare decision tree is sufficiently good. If at some step $j$, the optimal $f$-completion of decision tree $T^\circ$ has an error $\text{error}(T^*, f) \leq \frac{\epsilon}{2}$, then Lemma 5.4 (Case 1) guarantees that with high probability, $\widehat{\text{error}}_S(T) \leq \frac{3\epsilon}{4}$. This satisfies the termination condition, ensuring the algorithm makes progress and eventually halts.

### 5.4. Algorithm

The following algorithm formalizes the practical implementation of the top-down heuristic, incorporating the sample-based estimation procedures. It maintains separate sample sets for estimating scores, determining leaf labels, and estimating the final error, and it adaptively increases the size of these sets as the tree grows to maintain statistical confidence. A single iteration of this process is visualized in Figure 1. The algorithm is presented in Algorithm 2.

### 5.5. Runtime of the Practical Algorithm

To analyze the runtime of Algorithm 2, we begin by observing that each random sample can traverse at most $n$ internal nodes during the construction of the tree. Thus, the total runtime is proportional to the number of required samples multiplied by $n$.

Let $J$ denote the total number of splits performed by Algorithm 2. By Theorem 5.1, and aiming for $\epsilon/2$ error we have
$$
J \leq \max\left(\left(\frac{2e\cdot\Delta_{opt}}{\epsilon D_{opt}}\right)^{4\Delta_{opt}D_{opt}}, e^{4\Delta_{opt}D_{opt}}\right).
$$

Therefore, the total number of samples used depends on the bounds required for estimation accuracy at each split. Specifically, at each step $j$ we require $M_S(j, \delta, \epsilon/2, n)$ samples for score estimation, $M_{LL}(j, \epsilon/2, \delta)$ samples for leaf labeling, and $M_{EE}(j, \epsilon/2, \delta)$ samples for error estimation.

Summing over $j = 1, \ldots, J$ yields a total sample complexity of $O\left(J\log J \cdot n\log n \cdot \frac{1}{\epsilon^2} \cdot \log\frac{1}{\delta}\right)$. Since each sample may traverse up to $n$ nodes, the total runtime is larger by a factor of $n$, i.e. $O\left(J\log J \cdot n^2\log n \cdot \frac{1}{\epsilon^2} \cdot \log\frac{1}{\delta}\right)$.

This shows that even though the algorithm is parameter-free and sample-driven, its runtime remains polynomial in all relevant quantities for fixed $\Delta_{opt}$, $D_{opt}$, and $\epsilon$.

## 6. Conclusion

We extended the theoretical analysis of the classic top-down greedy decision tree heuristic from the uniform distribution to arbitrary product distributions. Our results show that if a target function admits an optimal decision tree with maximum depth $D_{opt}$ and average depth $\Delta_{opt}$ (un-

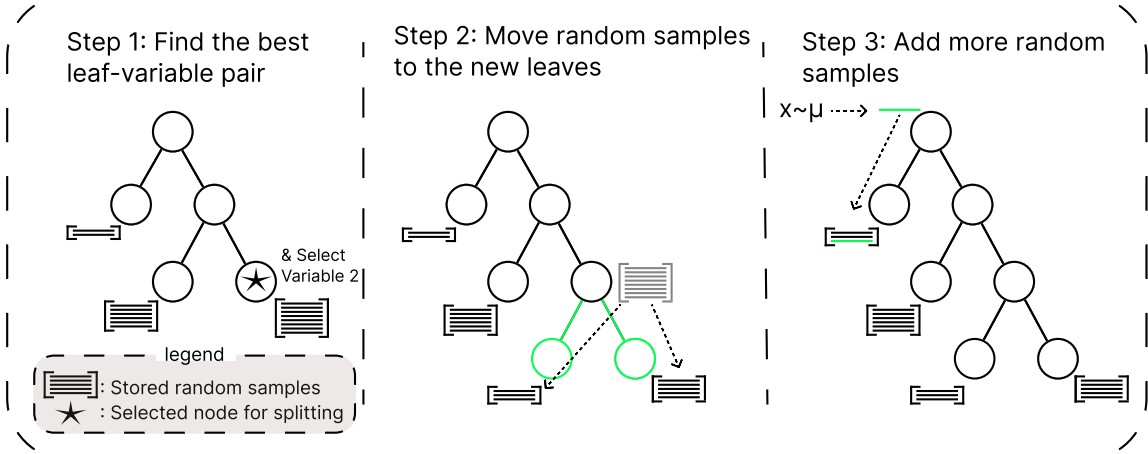

*Figure 1.* A visual depiction of one iteration of the practical top-down heuristic Algorithm 2. **Step 1:** The algorithm identifies the leaf and the variable that provide the best split based on estimated scores. **Step 2:** The samples stored at the selected leaf are partitioned between the two new children based on the splitting rule. **Step 3:** New random samples are drawn and added to the sample sets of all leaves in the tree to meet the increased sample complexity requirements for the next iteration.

der the product distribution), then the greedy heuristic constructs an $\epsilon$-approximating tree whose size is bounded by $\max\left(\left(\frac{e \cdot \Delta_{opt}}{\epsilon D_{opt}}\right)^{\Delta_{opt} D_{opt}}, e^{\Delta_{opt} D_{opt}}\right)$, with an analogous robustness guarantee when only approximate leaf scores are available. In addition, we gave a practical, parameter-free implementation based on sample estimates that does not require prior knowledge of the optimal tree size or depth, helping narrow the gap between provable guarantees and the greedy methods used in practice.

## 7. Limitations

Practical decision-tree implementations such as CART rely heavily on a post-hoc pruning step to mitigate overfitting, which can dramatically reduce the size of the final tree. We deliberately analyze the unpruned tree, both because rigorous theoretical analyses of pruning are still sparse and because, even for the pruned model, the pre-pruning size remains a meaningful quantity: it controls the worst-case runtime and memory cost of the growth phase, and (as our sample-complexity analysis in Section 5 shows) it controls the number of samples needed to maintain statistical confidence at each split. Extending our analysis to incorporate provable pruning rules is an interesting open direction.

## 8. Future Direction

A natural direction for future work is to extend the analysis beyond Boolean product spaces to continuous features and threshold-based splits, as used in CART-style implementations. Our proof relies on two ingredients tied to the discrete product-space setting: (i) the notion of *influence via coor-*

*dinate re-randomization*, and (ii) the structural inequality from (O'Donnell et al., 2005), which is established only for Boolean product spaces. By generalizing these two ingredients to threshold-based splits, we expect the analysis to plausibly carry over.

In addition Theorem 1.1 bounds the size of the greedy tree by a quantity exponential in $\Delta_{opt} D_{opt} \log(e/\epsilon)$. The super-polynomial lower bounds of Koch et al. (2023) imply that some quasi-polynomial dependence on $s$ is unavoidable, even with membership queries, so the exponent cannot be reduced to a constant. Whether the specific dependence on $\Delta_{opt} D_{opt}$ in the exponent is tight, or can be improved (for instance to $\Delta_{opt} + D_{opt}$, $\max(\Delta_{opt}, D_{opt})$, or to a quantity stated solely in $s$), is left as an open question.

## Impact Statement

This paper provides a better theoretical understanding of greedy decision tree learning algorithms beyond the uniform distribution, which may help increase the use of decision trees as a simple and easily interpretable decision-making model.

## Acknowledgements

This work is partially supported by DARPA expMath, ONR MURI 2024 award on Algorithms, Learning, and Game Theory, Army-Research Laboratory (ARL) grant W911NF2410052, NSF AF:Small grants 2218678, 2114269, 2347322.

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

## Appendix Roadmap

This appendix is organized as follows:

- Section B presents the full pseudocode for our practical, sample-based variant of the top-down heuristic.

- Section C collects proofs and supporting lemmas that were omitted from the main text for readability.

- Section D reports an empirical evaluation of the practical, parameter-free top-down heuristic (Algorithm 2).

- Section E works out two illustrative examples that quantify how the bound of Theorem 1.1 behaves in the balanced and the path-like (unbalanced) regimes.

## A. Extended related works

**Statistical-risk analyses of CART.** A complementary line of work studies the statistical generalization error of CART on samples drawn from a fixed (typically continuous) distribution. Klusowski & Tian (2024) prove that, as the sample size $N \to \infty$, the population risk of CART on additive regression models converges at a polynomial rate. Their results concern *statistical* risk decay with $N$ and address continuous features and pruning, which puts them in a different regime: they assume a prescribed data distribution and analyze how excess risk shrinks with more data, while we ask, given query access to a Boolean target $f$ admitting an optimal decision tree of small size and depth, whether the greedy heuristic itself produces a small tree close to $f$. Neither result implies the other.

**Parameterized complexity.** A separate body of work studies decision tree learning through the lens of parameterized complexity, where one is given an explicit dataset and seeks a tree of minimum size that classifies it correctly. Ordyniak & Szeider (2021) show that the problem is fixed-parameter tractable in the size of the optimal tree (assuming bounded domain size), but their algorithm is not top-down and is tailored to the explicitly-given-dataset model rather than the function-access model studied here. We view this line of work as orthogonal to ours.

## B. Practical Algorithm

We include here the full pseudocode for Algorithm 2.

## C. Omitted Proofs

We first state auxiliary lemmas used throughout the paper, and then give the deferred proofs in the order in which their statements appear in the main text.

### C.1. Auxiliary lemmas

To bound the number of steps, we must show that the Score of the chosen leaf is sufficiently large. We use the following critical inequality from (O'Donnell et al., 2005), which lower-bounds the maximal influence.

**Lemma C.1** (Max Influence vs. Variance, from (O'Donnell et al., 2005)). *For any function $f$ computable by a decision tree $T$, and any product distribution $\mu$:*

$$\max_i \text{Inf}_i(f) \geq \frac{\text{Var}(f)}{\Delta(T)}$$

*where $\Delta(T)$ is the average depth of $T$.*

#### C.1.1. VARIANCE BOUNDS

**Lemma C.2** (Variance Bounds). *For any Boolean function $f : \{0,1\}^n \to \{\pm 1\}$ and any $i \in [n]$, we have:*

$$\text{Inf}_i(f) \leq 2\text{error}(f, \pm 1) \leq \text{Var}(f)$$

---

**Algorithm 2** Practical Top-Down Heuristic for Decision Tree Induction

---

1: **function** PracticalBuildTopDownDT($f, \epsilon, \delta$)
2: Initialize $T^\circ$ with a single leaf, $v_{root}$. Let $j \leftarrow 1$.
3: For each leaf $v \in \text{leaves}(T^\circ)$, initialize sample sets:
4: $S_v^{LL} \leftarrow$ Draw $M_{LL}(j, \epsilon, \delta)$ samples. {For leaf labeling}
5: $S_v^{EE} \leftarrow$ Draw $M_{EE}(j, \epsilon, \delta)$ samples. {For error estimation}
6: For $i = 1 \ldots n$, $E_v^{(i)} \leftarrow$ Draw $M_S(j, \delta, \epsilon, n)$ sample pairs. {For score estimation}
7: **while** True **do**
8:     *// Estimate scores, labels, and error for all leaves*
9:     **for** each leaf $v \in \text{leaves}(T^\circ)$ **do**
10:         For $i = 1 \ldots n$, calculate $\widehat{\text{Score}}(v, i, E_v^{(i)})$ using Equation (1).
11:         $label_v \leftarrow$ majority label of $f(x)$ for $x \in S_v^{LL}$.
12:         $mc_v \leftarrow \#\{x \in S_v^{EE} \mid f(x) \neq label_v\}$.
13:     **end for**
14:     *// Check termination condition*
15:     **if** $\sum_{v \in \text{leaves}(T^\circ)} mc_v \leq \frac{3\epsilon}{4} \cdot M_{EE}(j, \epsilon, \delta)$ **then**
16:         **return** $T^\circ$ with labels $\{label_v\}$.
17:     **end if**
18:     $j \leftarrow j + 1$.
19:     *// Select best leaf and split*
20:     $v^*, i^* \leftarrow \arg\max_{v,i} \widehat{\text{Score}}(v, i, E_v^{(i)})$.
21:     Split leaf $v^*$ on variable $i^*$ into new children $v_0, v_1$.
22:     *// Partition old samples and draw new ones to meet new sample complexity*
23:     **for** each new leaf $v_{new} \in \{v_0, v_1\}$ **do**
24:         Partition samples from $S_{v^*}^{LL}, S_{v^*}^{EE}$ and $E_{v^*}^{(i)}$ that reach $v_{new}$ into new sets $S_{v_{new}}^{LL}, S_{v_{new}}^{EE}, E_{v_{new}}^{(i)}$.
25:     **end for**
26:     **for** each leaf $v \in \text{leaves}(T^\circ)$ **do**
27:         Draw $M_{LL}(j + 1, \epsilon, \delta) - M_{LL}(j, \epsilon, \delta)$ new samples and add to $S_v^{LL}$ for the $v$ they reach.
28:         Draw $M_{EE}(j + 1, \epsilon, \delta) - M_{EE}(j, \epsilon, \delta)$ new samples and add to $S_v^{EE}$ for the $v$ they reach.
29:         **for** $i = 1 \ldots n$ **do**
30:             Draw $M_S(j + 1, \delta, \epsilon, n) - M_S(j, \delta, \epsilon, n)$ new sample pairs and add to $E_v^{(i)}$ for the $v$ they reach.
31:         **end for**
32:     **end for**
33: **end while**

---

*Proof of Lemma C.2.* First we prove that $\text{Var}(f) \geq 2\text{error}(f, \pm 1)$. W.L.O.G. assume $\mu = \Pr[f(x) = +1] \leq \Pr[f(x) = -1]$, so $\text{error}(f, \pm 1) = \mu$ and $\mu < \frac{1}{2}$. Then $\text{Var}(f) = 1 - (2\mu - 1)^2 = 4\mu(1 - \mu) > 2\mu = 2\text{error}(f, \pm 1)$. Moreover, we know:

$$\text{Inf}_i(f) = \Pr[f(x) \neq f(x^{(i)})] \leq \Pr[f(x) \neq f(x^{\oplus i})]$$
$$= 2\Pr[f(x) = +1, f(x^{\oplus i}) = -1] \leq 2\Pr[f(x) = +1] = 2\mu = 2\text{error}(f, \pm 1)$$

$\square$

## C.2. Proof of Lemma 4.1 (Error is Bounded by Cost)

**Lemma 4.1** (Error is Bounded by Cost). *For any bare tree $T^\circ$ and function $f$, the error of its $f$-completion is bounded by its cost:*

$$\text{error}(T^\circ, f) \leq \text{cost}(T^\circ)$$

*Proof of Lemma 4.1.* We decompose the error of the unlabeled tree $T^\circ$ by leaves. For each leaf $v$, by definition we know:

$$\text{error}(f_v, \pm 1) := \min\left(\text{error}(f_v, -1), \text{error}(f_v, 1)\right),$$

so that

$$\text{error}(T^\circ, f) = \sum_{v \in \text{leaves}(T^\circ)} p_v \cdot \text{error}(f_v, \pm 1).$$

It's enough to prove $\text{error}(f_v, \pm 1) \leq \text{Inf}(f_v)$ for every leaf $v$.

**Base case** ($n = 1$): Let $f$ depend only on one variable $x_1$. If $f(0) = f(1)$, then both the error and influence are 0. Otherwise, assume w.l.o.g. that $0 < \mu_1 \leq \frac{1}{2}$. Then:

$$\text{error}(f_v, \pm 1) = \mu_1, \quad \text{Inf}(f_v) = 4\mu_1(1 - \mu_1).$$

We compute:

$$4\mu_1(1 - \mu_1) - \mu_1 = 3\mu_1 - 4\mu_1^2 > 0,$$

so $\text{error}(f_v, \pm 1) \leq \text{Inf}(f_v)$. A symmetric argument applies when $\mu_1 > \frac{1}{2}$.

**Inductive step:** Assume the inequality holds for all functions on $n - 1$ variables. Let $f$ be a function on $n$ variables, and define $f_0 := f_{x_n=0}$ and $f_1 := f_{x_n=1}$. Let $x := \Pr[f_0 = -1]$, $y := \Pr[f_1 = -1]$. Assume w.l.o.g. that $x \geq \frac{1}{2}$, $y \leq \frac{1}{2}$, and the overall majority label is $+1$.

Then:

$$\mu_n x + (1 - \mu_n)y \leq \mu_n(1 - x) + (1 - \mu_n)(1 - y) \Rightarrow y \leq \frac{\frac{1}{2} - \mu_n x}{1 - \mu_n}.$$

By the inductive hypothesis: $1 - x \leq \text{cost}_l$, $y \leq \text{cost}_r$. Now:

$$4\left(\frac{1}{2} - \mu_n x\right) \leq \text{cost}_l - x + 4(1 - \mu_n)x,$$

and thus:

$$4(1 - \mu_n)y \leq \text{cost}_l - x + 4(1 - \mu_n)x.$$

Multiply both sides by $\mu_n$:

$$4\mu_n(1 - \mu_n)y \leq \mu_n(\text{cost}_l - x + 4(1 - \mu_n)x).$$

Also, from $y \leq \text{cost}_r$, we get:

$$(1 - \mu_n)y \leq (1 - \mu_n)\text{cost}_r.$$

Summing both:

$$4\mu_n(1 - \mu_n)y + (1 - \mu_n)y \leq \mu_n(\text{cost}_l - x + 4(1 - \mu_n)x) + (1 - \mu_n)\text{cost}_r.$$

Hence,

$$\mu_n x + (1 - \mu_n)y \leq \mu_n \text{cost}_l + (1 - \mu_n)\text{cost}_r + 4\mu_n(1 - \mu_n)(x - y),$$

which implies:
$$\text{error}(f_v, \pm 1) \le \text{Inf}(f_v).$$

Thus, by induction, for all $v$,

$$\text{error}(f_v, \pm 1) \le \text{Inf}(f_v) \quad \Rightarrow \quad \text{error}(T^\circ, f) \le \text{cost}(T^\circ).$$

$\square$

## C.3. Proof of Lemma 4.2 (Total Influence vs. Variance and Depth)

**Lemma 4.2** (Total Influence vs. Variance and Depth). *For any function $f : \{0,1\}^n \to \{\pm 1\}$ represented by a decision tree $T$ with depth $D(T)$, we have:*
$$\text{Inf}(f) \le D(T) \cdot \text{Var}(f) \quad \text{and} \quad \text{Inf}(f) \le \Delta(T)$$

*Proof of Lemma 4.2.* We begin by expanding the total influence:

$$\text{Inf}(f) = \sum_i \text{Inf}_i(f) = \sum_i \mathbb{E}_{x \sim \mu}[f(x) \ne f(x^{(i)})].$$

If for some $x$ and coordinate $i$, we have $f(x) \ne f(x^{(i)})$, then in the decision tree $T$, there must exist a node $v$ that queries coordinate $i$, and where changing $x_i$ changes the output. The contribution of such a node is $\text{Inf}_{i(v)}(f_v) \cdot p_v$, so:

$$\text{Inf}(f) = \sum_v \text{Inf}_{i(v)}(f_v) \cdot p_v.$$

Using Lemma C.2, $\text{Inf}_{i(v)}(f_v) \le \text{Var}(f_v)$, we get:

$$\text{Inf}(f) \le \sum_v \text{Var}(f_v) \cdot p_v = \sum_{d=1}^{D(T)} \sum_{v:\text{depth}(v)=d} \text{Var}(f_v) \cdot p_v.$$

At each depth $d$, the subfunctions $f_v$ form a partition of $f$, so by the law of total variance:

$$\sum_{v:\text{depth}(v)=d} \text{Var}(f_v) \cdot p_v \le \text{Var}(f) \quad \Rightarrow \quad \text{Inf}(f) \le D(T) \cdot \text{Var}(f).$$

Moreover, since $\text{Inf}_{i(v)}(f_v) \le 1$ for all $v$, we also have:

$$\text{Inf}(f) = \sum_v \text{Inf}_{i(v)}(f_v) \cdot p_v \le \sum_v p_v = \Delta.$$

$\square$

## C.4. Proof of Lemma 4.3 (Cost Reduction per Step)

**Lemma 4.3** (Cost Reduction per Step). *Let $T^\circ$ be a bare tree, and let $T'^\circ$ be the tree obtained by splitting a leaf $v \in \text{leaves}(T^\circ)$ on its most influential variable, $i(v)$. The cost of the new tree is:*

$$\text{cost}(T'^\circ) = \text{cost}(T^\circ) - \text{Score}(v)$$

*Proof of Lemma 4.3.* The cost of a tree is the sum of costs contributed by its leaves. When we split leaf $v$ into new leaves $v_0$ and $v_1$, the leaves in $\text{leaves}(T^\circ) \setminus \{v\}$ are unaffected. Thus, the change in cost is:

$$\text{cost}(T'^\circ) - \text{cost}(T^\circ) = (p_{v_0} \text{Inf}(f_{v_0}) + p_{v_1} \text{Inf}(f_{v_1})) - p_v \text{Inf}(f_v)$$

Let $\mu_{i(v)} = \Pr[x_{i(v)} = 1]$. Then $p_{v_0} = p_v(1 - \mu_{i(v)})$ and $p_{v_1} = p_v\mu_{i(v)}$. Substituting gives:

$$\text{cost}(T'^{\circ}) - \text{cost}(T^{\circ}) = p_v\left[(1 - \mu_{i(v)})\text{Inf}(f_{v_0}) + \mu_{i(v)}\text{Inf}(f_{v_1}) - \text{Inf}(f_v)\right] \tag{2}$$

We relate the total influence of $f_v$ to its subfunctions. The influence of any variable $k \neq i(v)$ on $f_v$ is the weighted average of its influence on the subfunctions: $\text{Inf}_k(f_v) = (1 - \mu_{i(v)})\text{Inf}_k(f_{v_0}) + \mu_{i(v)}\text{Inf}_k(f_{v_1})$. The influence of the splitting variable $i(v)$ on the subfunctions $f_{v_0}$ and $f_{v_1}$ is zero, as it is fixed. The total influence of $f_v$ can be decomposed as:

$$\begin{aligned}
\text{Inf}(f_v) &= \text{Inf}_{i(v)}(f_v) + \sum_{k \neq i(v)} \text{Inf}_k(f_v) \\
&= \text{Inf}_{i(v)}(f_v) + \sum_{k \neq i(v)} \left[(1 - \mu_{i(v)})\text{Inf}_k(f_{v_0}) + \mu_{i(v)}\text{Inf}_k(f_{v_1})\right] \\
&= \text{Inf}_{i(v)}(f_v) + (1 - \mu_{i(v)})\text{Inf}(f_{v_0}) + \mu_{i(v)}\text{Inf}(f_{v_1})
\end{aligned}$$

Substituting this back into the expression for Equation (2)

$$\begin{aligned}
\text{cost}(T'^{\circ}) - \text{cost}(T^{\circ}) &= p_v\left[(1 - \mu_{i(v)})\text{Inf}(f_{v_0}) + \mu_{i(v)}\text{Inf}(f_{v_1}) - \text{Inf}(f_v)\right] \\
&= p_v\Big[(1 - \mu_{i(v)})\text{Inf}(f_{v_0}) + \mu_{i(v)}\text{Inf}(f_{v_1}) \\
&\qquad - \big(\text{Inf}_{i(v)}(f_v) + (1 - \mu_{i(v)})\text{Inf}(f_{v_0}) + \mu_{i(v)}\text{Inf}(f_{v_1})\big)\Big] \\
&= -p_v \, \text{Inf}_{i(v)}(f_v) = -\text{Score}(v)
\end{aligned}$$

Hence we have

$$\text{cost}(T'^{\circ}) = \text{cost}(T^{\circ}) - \text{Score}(v).$$

$\square$

## C.5. Proof of Lemma 4.6 (Phase 1: High Cost Reduction)

**Lemma 4.6** (Phase 1: High Cost Reduction). *Let $J_1$ be the number of steps required for the cost $C_j$ to drop from its initial value to at most $\epsilon D_{opt}$ in Algorithm 1. Then:*

$$J_1 \leq \max\left(\left(\frac{\Delta_{opt}}{\epsilon D_{opt}}\right)^{D_{opt}\Delta_{opt}}, 1\right)$$

*Proof of Lemma 4.6.* During Phase 1 we have $C_j > \epsilon D_{opt}$. In this regime, Lemma 4.5 yields the one-step progress bound

$$C_{j+1} = C_j - \text{Score}(l^*) \leq C_j - \frac{C_j}{j\, D_{opt}\, \Delta_{opt}} = C_j\left(1 - \frac{1}{j\, D_{opt}\, \Delta_{opt}}\right).$$

Fix any integer $t \geq 1$. Iterating this inequality for $j = 1, 2, \ldots, t$ gives

$$C_{t+1} \leq C_1 \prod_{j=1}^{t}\left(1 - \frac{1}{j\, D_{opt}\, \Delta_{opt}}\right).$$

Taking logs and using $\ln(1 - x) \leq -x$ for $x \in (0, 1)$, we obtain

$$\begin{aligned}
C_{t+1} &\leq C_1 \exp\left(\sum_{j=1}^{t} \ln\left(1 - \frac{1}{j\, D_{opt}\, \Delta_{opt}}\right)\right) \\
&\leq C_1 \exp\left(-\sum_{j=1}^{t} \frac{1}{j\, D_{opt}\, \Delta_{opt}}\right) = C_1 \exp\left(-\frac{1}{D_{opt}\, \Delta_{opt}}\sum_{j=1}^{t} \frac{1}{j}\right).
\end{aligned}$$

Lower bounding the harmonic sum by $\sum_{j=1}^{t} \frac{1}{j} \geq \ln(t)$ yields

$$C_{t+1} \leq C_1 \exp\left(-\frac{\ln(t)}{D_{opt}\,\Delta_{opt}}\right) = C_1\, t^{-1/(D_{opt}\,\Delta_{opt})}.$$

Finally, using $C_1 = \mathrm{Inf}(f) \leq \Delta_{opt}$ (Lemma 4.2) we get the uniform bound

$$C_{t+1} \leq \Delta_{opt}\, t^{-1/(D_{opt}\,\Delta_{opt})}.$$

Now choose

$$T := \left(\frac{\Delta_{opt}}{\epsilon D_{opt}}\right)^{D_{opt}\Delta_{opt}}.$$

For any integer $t \geq T$ we have

$$\Delta_{opt}\, t^{-1/(D_{opt}\Delta_{opt})} \leq \Delta_{opt}\, T^{-1/(D_{opt}\Delta_{opt})} = \epsilon D_{opt},$$

and therefore $C_{t+1} \leq \epsilon D_{opt}$.

Let $J_1$ be the first index such that $C_{J_1+1} \leq \epsilon D_{opt}$. Since $C_{t+1} \leq \epsilon D_{opt}$ holds for every integer $t \geq T$. By minimality of $J_1$ we conclude

$$J_1 \leq \max\left(\left(\frac{\Delta_{opt}}{\epsilon D_{opt}}\right)^{D_{opt}\Delta_{opt}}, 1\right).$$

$\square$

### C.6. Proof of Lemma 4.7 (Phase 2: Termination)

**Lemma 4.7** (Phase 2: Termination). *Let $J_1$ be the step at which the cost drops to $\epsilon D_{opt}$. The Algorithm 1 guarantees termination at a total number of steps $J_{total}$ satisfying:*

$$J_{total} \leq J_1 \cdot e^{\Delta_{opt} D_{opt}}$$

*Proof of Lemma 4.7.* Consider the steps where $\epsilon < \mathrm{error}(T_j^\circ) \leq C_j \leq \epsilon D_{opt}$. We use the bound from Lemma 4.4. The cost reduction at step $j$ (where $j > J_1$) is at least $\mathrm{Score}(l^*) \geq \frac{\epsilon}{j \cdot \Delta_{opt}}$. Let $J_2$ be the number of steps in this phase. The total cost reduction required is at most $C_{J_1+1} \leq \epsilon D_{opt}$. The total reduction is $\sum_{j=J_1}^{J_1+J_2} \mathrm{Score}(l_j^*) \geq \sum_{j=J_1}^{J_1+J_2} \frac{\epsilon}{j \cdot \Delta_{opt}}$. We require:

$$\frac{\epsilon}{\Delta_{opt}} \sum_{j=J_1}^{J_1+J_2} \frac{1}{j} \geq \epsilon D_{opt} \implies \sum_{j=J_1}^{J_{total}} \frac{1}{j} \geq \Delta_{opt} D_{opt}$$

Using the integral bound $\sum_{k=a}^{b} \frac{1}{k} > \ln(\frac{b+1}{a})$, a sufficient condition for termination is:

$$\ln\left(\frac{J_{total}+1}{J_1}\right) \geq \Delta_{opt} D_{opt} \implies J_{total} + 1 \geq J_1 e^{\Delta_{opt} D_{opt}}$$

This implies $J_{total} \leq J_1 e^{\Delta_{opt} D_{opt}}$. $\square$

### C.7. Proof of Theorem 5.1 (Robust Performance with Approximate Scores)

**Theorem 5.1** (Robust Performance with Approximate Scores). *Let the conditions of Theorem 1.1 hold. If at each step the algorithm selects a leaf $l'$ to split such that $\mathrm{Score}(l') \geq \frac{1}{4} \max_l \mathrm{Score}(l)$, then it returns an $\epsilon$-approximating tree of size at most:*

$$\max\left(\left(\frac{e \cdot \Delta_{opt}}{\epsilon D_{opt}}\right)^{4\Delta_{opt} D_{opt}}, e^{4\Delta_{opt} D_{opt}}\right)$$

*Proof of Theorem 5.1.* We follow exactly the same potential-function analysis used to prove Theorem 1.1. Let $T_j^\circ$ denote the bare tree after $j - 1$ splits, and let $C_j := \text{cost}(T_j^\circ)$ be its cost.

As in the proof of Theorem 1.1, Lemma 4.1 gives $\text{error}(T_j^\circ, f) \leq C_j$, so the algorithm must terminate once $C_j \leq \epsilon$. Also, Lemma 4.2 gives the initial bound $C_1 = \text{Inf}(f) \leq \Delta_{opt}$.

The only change is that the leaf chosen in step $j$ is not necessarily the maximizer of $\text{Score}(\cdot)$, but satisfies

$$\text{Score}(l_j') \geq \frac{1}{4} \max_{l \in \text{leaves}(T_j^\circ)} \text{Score}(l).$$

Therefore, in any inequality of the form $\max_l \text{Score}(l) \geq$ (lower bound), we can replace it by $\text{Score}(l_j') \geq \frac{1}{4} \cdot$ (lower bound). We now redo the two phases with this loss.

**Phase 1: high cost.** Assume $C_j > \epsilon D_{opt}$. By Lemma 4.5, in this regime

$$\max_l \text{Score}(l) \geq \frac{C_j}{j \, D_{opt} \, \Delta_{opt}}.$$

Hence

$$\text{Score}(l_j') \geq \frac{1}{4} \cdot \frac{C_j}{j \, D_{opt} \, \Delta_{opt}}.$$

By Lemma 4.3, the cost drops by the score of the split leaf, so

$$C_{j+1} = C_j - \text{Score}(l_j') \leq C_j \left( 1 - \frac{1}{4j \, D_{opt} \, \Delta_{opt}} \right).$$

Iterating as in the proof of Lemma 4.6 (using $\ln(1 - x) \leq -x$ and a harmonic-series lower bound) yields that for every integer $t \geq 1$,

$$C_{t+1} \leq C_1 \, t^{-1/(4D_{opt}\Delta_{opt})} \leq \Delta_{opt} \, t^{-1/(4D_{opt}\Delta_{opt})}.$$

In particular, choosing

$$T_1 := \left( \frac{\Delta_{opt}}{\epsilon D_{opt}} \right)^{4D_{opt}\Delta_{opt}},$$

we get that for all $t \geq T_1$, $C_{t+1} \leq \epsilon D_{opt}$. Let $J_1$ be the first index such that $C_{J_1+1} \leq \epsilon D_{opt}$. Then

$$J_1 \leq \max \left( \left( \frac{\Delta_{opt}}{\epsilon D_{opt}} \right)^{4D_{opt}\Delta_{opt}}, 1 \right).$$

**Phase 2: termination once error remains high.** Now consider steps with $\epsilon < \text{error}(T_j^\circ, f) \leq C_j \leq \epsilon D_{opt}$. By Lemma 4.4, for such a step we have

$$\max_l \text{Score}(l) \geq \frac{2\epsilon}{j \, \Delta_{opt}}.$$

Thus

$$\text{Score}(l_j') \geq \frac{1}{4} \cdot \frac{2\epsilon}{j \, \Delta_{opt}} = \frac{\epsilon}{2j \, \Delta_{opt}}.$$

Again using Lemma 4.3, we get

$$C_{j+1} \leq C_j - \frac{\epsilon}{2j \, \Delta_{opt}}.$$

**Concluding Phase 2.** Unrolling the above recurrence from step $J_1 + 1$ up to some step $J \geq J_1 + 1$ gives

$$C_{J+1} \leq C_{J_1+1} - \frac{\epsilon}{2\Delta_{opt}} \sum_{j=J_1+1}^{J} \frac{1}{j}.$$

Since we enter Phase 2 with $C_{J_1+1} \leq \epsilon D_{opt}$, it suffices to ensure that

$$\frac{\epsilon}{2\Delta_{opt}} \sum_{j=J_1+1}^{J} \frac{1}{j} \geq \epsilon D_{opt},$$

i.e.,

$$\sum_{j=J_1+1}^{J} \frac{1}{j} \geq 2\Delta_{opt} D_{opt}.$$

Using the standard harmonic-series bound $\sum_{j=a}^{b} \frac{1}{j} \geq \ln\left(\frac{b+1}{a}\right)$, a sufficient condition is

$$\ln\left(\frac{J+1}{J_1+1}\right) \geq 2\Delta_{opt} D_{opt} \implies J+1 \geq (J_1+1)e^{2\Delta_{opt}D_{opt}}.$$

Therefore, the algorithm must terminate by

$$J_{total} \leq (J_1+1)e^{2\Delta_{opt}D_{opt}}.$$

Finally, using the Phase 1 bound $J_1 \leq \max\{(\Delta_{opt}/(\epsilon D_{opt}))^{4\Delta_{opt}D_{opt}}, 1\}$ and the crude inequality $(J_1+1)e^{2\Delta_{opt}D_{opt}} \leq J_1 e^{4\Delta_{opt}D_{opt}}$, we get

$$J_{total} \leq \max\left(\left(\frac{e \cdot \Delta_{opt}}{\epsilon D_{opt}}\right)^{4\Delta_{opt}D_{opt}}, e^{4\Delta_{opt}D_{opt}}\right),$$

which is exactly the claimed size bound (up to the final $+1$ leaf). $\qquad\square$

### C.8. Proof of Lemma 5.2 (Unbiasedness of the Score Estimator)

**Lemma 5.2** (Unbiasedness of the Score Estimator). *The estimator* $\widehat{\text{Score}}(l, i, E_i)$ *is an unbiased estimator of the true score* Score$(l, i)$.

*Proof of Lemma 5.2.* We show that $\widehat{\text{Score}}(l, i, E_i)$ is an unbiased estimator of Score$(l, i) = \Pr[x \text{ reaches } l] \cdot \text{Inf}_i(f_l)$. By definition:

$$\widehat{\text{Score}}(l, i, E_i) = \frac{1}{|E_i|} \sum_{(x,x^{(i)})\in E_i} \mathbf{1}[x \text{ and } x^{(i)} \text{ reach } l] \cdot \mathbf{1}[f(x) \neq f(x^{(i)})].$$

Taking expectation:

$$\mathbb{E}[\widehat{\text{Score}}(l, i, E_i)] = \Pr[x, x^{(i)} \text{ reach } l] \cdot \Pr[f(x) \neq f(x^{(i)}) \mid x, x^{(i)} \text{ reach } l].$$

Since $i$ is not fixed in the path to $l$, $x$ reaches $l$ iff $x^{(i)}$ does, so:

$$\Pr[x, x^{(i)} \text{ reach } l] = \Pr[x \text{ reaches } l] = p_l,$$

and

$$\Pr[f(x) \neq f(x^{(i)}) \mid x, x^{(i)} \text{ reach } l] = \text{Inf}_i(f_l).$$

Thus:

$$\mathbb{E}[\widehat{\text{Score}}(l, i, E_i)] = p_l \cdot \text{Inf}_i(f_l) = \text{Score}(l, i).$$

$$\square$$

## C.9. Proof of Lemma 5.3

**Lemma 5.3.** *Let $T^\circ$ be a decision tree with $l$ leaves. Let $T$ be the tree obtained by assigning a label from $\{-1,+1\}$ to each leaf of $T^\circ$ based on the majority vote of $\hat{m}_j$ samples drawn i.i.d. from a distribution $D$ and labeled by a function $f$. Let $T^*$ be the optimal labeling of $T^\circ$. For any $\delta \in (0,1)$, with probability at least $1 - \frac{\delta}{8j^2}$, the following bound holds:*

$$|\text{error}(T,f) - \text{error}(T^*,f)| \leq \sqrt{\frac{2(l\ln(2) + \ln(16j^2/\delta))}{\hat{m}_j}}$$

*Proof of Theorem 5.3.* Let $\mathcal{H}$ be the hypothesis space of all possible labelings of the $l$ leaves of $T^\circ$. The size of this space is $|\mathcal{H}| = 2^l$. Let $S$ be the set of $\hat{m}_j$ samples. For any hypothesis $h \in \mathcal{H}$, we denote its true error by $\text{error}(h,f)$ and its empirical error on $S$ by $\widehat{\text{error}}_S(h)$.

The tree $T$ is the result of the Empirical Risk Minimization (ERM) principle, meaning $T = \arg\min_{h \in \mathcal{H}} \widehat{\text{error}}_S(h)$. The tree $T^*$ is the true error minimizer, $T^* = \arg\min_{h \in \mathcal{H}} \text{error}(h,f)$.

We use a standard uniform convergence bound for finite hypothesis spaces. For a chosen confidence parameter $\delta' \in (0,1)$, with probability at least $1 - \delta'$, the following holds for all $h \in \mathcal{H}$ simultaneously:

$$|\text{error}(h,f) - \widehat{\text{error}}_S(h)| \leq \sqrt{\frac{\ln|\mathcal{H}| + \ln(2/\delta')}{2\hat{m}_j}}$$

Let $\epsilon_{\text{uc}} = \sqrt{\frac{\ln|\mathcal{H}| + \ln(2/\delta')}{2\hat{m}_j}}$. Assuming this high-probability event occurs, we can bound the true error of $T$:

$$
\begin{aligned}
\text{error}(T,f) &\leq \widehat{\text{error}}_S(T) + \epsilon_{\text{uc}} && \text{(by the uniform convergence bound)} \\
&\leq \widehat{\text{error}}_S(T^*) + \epsilon_{\text{uc}} && \text{(since } T \text{ is the ERM hypothesis, } \widehat{\text{error}}_S(T) \leq \widehat{\text{error}}_S(T^*)) \\
&\leq \text{error}(T^*,f) + \epsilon_{\text{uc}} + \epsilon_{\text{uc}} && \text{(by the uniform convergence bound on } T^*) \\
&= \text{error}(T^*,f) + 2\epsilon_{\text{uc}}
\end{aligned}
$$

This gives the excess error bound $\text{error}(T,f) - \text{error}(T^*,f) \leq 2\epsilon_{\text{uc}}$. By definition of $T^*$ as the optimal hypothesis, $\text{error}(T,f) \geq \text{error}(T^*,f)$, so the difference is non-negative. Therefore, $|\text{error}(T,f) - \text{error}(T^*,f)| = \text{error}(T,f) - \text{error}(T^*,f)$.

We set our confidence parameter $\delta' = \frac{\delta}{8j^2}$. Substituting this and $|\mathcal{H}| = 2^l$ into the excess error bound gives:

$$
\begin{aligned}
|\text{error}(T,f) - \text{error}(T^*,f)| &\leq 2\sqrt{\frac{l\ln 2 + \ln(2/\delta')}{2\hat{m}_j}} = \sqrt{\frac{4(l\ln 2 + \ln(16j^2/\delta))}{2\hat{m}_j}} \\
&= \sqrt{\frac{2(l\ln 2 + \ln(16j^2/\delta))}{\hat{m}_j}}
\end{aligned}
$$

This bound holds with probability at least $1 - \delta' = 1 - \frac{\delta}{8j^2}$, which completes the proof. $\square$

## C.10. Proof of Lemma 5.4 (Reliable Error Estimation)

**Lemma 5.4** (Reliable Error Estimation). *Let $\epsilon > 0$ be a target accuracy. Let $T$ be the tree from Part 1, constructed with $\hat{m}_j$ samples such that $\sqrt{\frac{2(l\ln(2) + \ln(16j^2/\delta))}{\hat{m}_j}} \leq \frac{\epsilon}{8}$. Let $S$ be a new set of $m_j''$ i.i.d. samples, where $m_j'' \geq \frac{32}{\epsilon^2}\ln(\frac{16j^2}{\delta})$. Let the event in Lemma 5.3 hold. Then:*

$$
\begin{cases}
\text{If } \text{error}(T^*,f) \leq \frac{\epsilon}{2} & \Pr\left[\widehat{\text{error}}_S(T) > \frac{3\epsilon}{4}\right] \leq \frac{\delta}{8j^2}. \\
\text{If } \text{error}(T,f) > \epsilon & \Pr\left[\widehat{\text{error}}_S(T) \leq \frac{3\epsilon}{4}\right] \leq \frac{\delta}{8j^2}.
\end{cases}
$$

*Proof of Lemma 5.4.* We prove each case separately, assuming the high-probability event from Lemma 5.3 holds. Let $\epsilon_1 = |\text{error}(T, f) - \text{error}(T^*, f)|$. From the condition on $\hat{m}_j$ and Lemma 5.3, we have $\epsilon_1 \leq \frac{\epsilon}{8}$.

**Case 1: Assume** $\text{error}(T^*, f) \leq \frac{\epsilon}{2}$. Our goal is to show that the empirical error is unlikely to be high. First, we bound the true error of our tree $T$:

$$\text{error}(T, f) \leq \text{error}(T^*, f) + \epsilon_1 \leq \frac{\epsilon}{2} + \frac{\epsilon}{8} = \frac{5\epsilon}{8}$$

We want to bound the probability $\Pr[\widehat{\text{error}}_S(T) > \frac{3\epsilon}{4}]$. This event requires the empirical error to deviate from its mean, $\text{error}(T, f)$, by at least $\frac{3\epsilon}{4} - \text{error}(T, f) \geq \frac{3\epsilon}{4} - \frac{5\epsilon}{8} = \frac{\epsilon}{8}$. By a one-sided Hoeffding's inequality:

$$\Pr\left[\widehat{\text{error}}_S(T) > \frac{3\epsilon}{4}\right] \leq \Pr\left[\widehat{\text{error}}_S(T) - \text{error}(T, f) \geq \frac{\epsilon}{8}\right]$$
$$\leq e^{-2m_j''(\epsilon/8)^2} = e^{-m_j''\epsilon^2/32}$$

Given $m_j'' \geq \frac{32}{\epsilon^2} \ln(\frac{16j^2}{\delta})$, this probability is bounded by $\frac{\delta}{16j^2}$, which is less than the required $\frac{\delta}{8j^2}$.

**Case 2: Assume** $\text{error}(T, f) > \epsilon$. Our goal is to show that the empirical error is unlikely to be low. We want to bound the probability $\Pr[\widehat{\text{error}}_S(T) \leq \frac{3\epsilon}{4}]$. This event requires the empirical error to be smaller than its mean by at least $\text{error}(T, f) - \frac{3\epsilon}{4} > \epsilon - \frac{3\epsilon}{4} = \frac{\epsilon}{4}$. By a one-sided Hoeffding's inequality:

$$\Pr\left[\widehat{\text{error}}_S(T) \leq \frac{3\epsilon}{4}\right] \leq \Pr\left[\text{error}(T, f) - \widehat{\text{error}}_S(T) \geq \frac{\epsilon}{4}\right]$$
$$\leq e^{-2m_j''(\epsilon/4)^2} = e^{-m_j''\epsilon^2/8}$$

Given our condition on $m_j''$, we have $e^{-m_j''\epsilon^2/8} \leq e^{-4\ln(16j^2/\delta)} \ll \frac{\delta}{8j^2}$. $\qquad\square$

# D. Empirical Evaluation

In this section we empirically evaluate the practical, parameter-free top-down heuristic (Algorithm 2). The implementation matches the pseudocode of Algorithm 2: at each step it estimates the score $\text{Score}(l, i)$ from sample pairs (Equation (1)), assigns leaf labels by majority vote on a separate sample set, and tests termination by estimating the empirical error of the labeled tree on a third independent sample set, augmenting all sample sets adaptively as the tree grows. All experiments are reported as the mean across 6 independent repetitions, with error bars showing $\pm 1$ standard deviation.

**Setup.** We construct ground-truth Boolean targets $f : \{0, 1\}^n \to \{\pm 1\}$ from two structures: a fully balanced decision tree (*balanced DT*) and a chain-like decision tree (*unbalanced DT*). Inputs are drawn from a product distribution where each coordinate is independently 1 with probability $p$ and 0 with probability $1 - p$; we sweep $p \in \{0.5, 0.3, 0.1\}$, ranging from the uniform distribution to a heavily biased one.

**Tree size vs. $\epsilon$ (Figure 2a).** We fix $n = 20$, $\delta = 0.1$, and run Algorithm 2 for $\epsilon \in \{0.10, 0.15, 0.20, 0.25, 0.30\}$. The targets are a balanced DT of depth 4 (16 leaves) and an unbalanced DT with 16 leaves. Figure 2(a) plots the size of the recovered tree against $1/\epsilon$ on a log–log scale. The recovered trees stay close in size to the ground-truth target across all configurations, in line with the size bound of Theorem 5.1.

**Tree size vs. dimension (Figure 2b).** We fix $\epsilon = 0.15$, $\delta = 0.1$, and sweep $n \in \{3, 4, 5, 6, 7\}$. The targets are a balanced DT of depth 3 (8 leaves) and an unbalanced DT with 8 leaves. Figure 2(b) reports the size of the recovered tree as $n$ grows. As predicted by Theorem 1.1, the size grows mildly with $n$ across all three product distributions, with no exponential blow-up; on the unbalanced target, the size remains close to the ground-truth 8 leaves even as $n$ varies.

**Takeaways.** Across all configurations, the recovered tree stays close in size to the ground-truth target and well within the size bound of Theorem 5.1, and the sample complexity grows polynomially in $n$ and $1/\epsilon$, in agreement with the runtime analysis of Section 5.

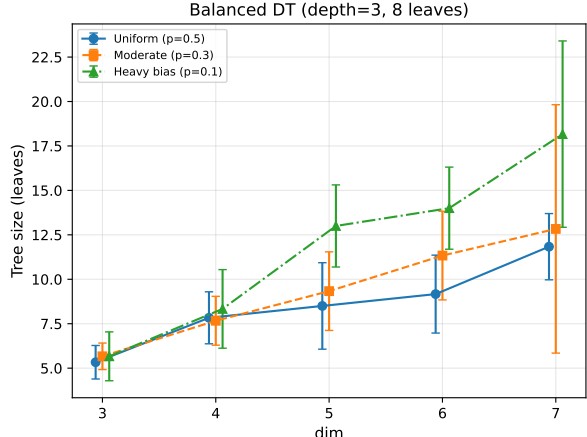

*(a)* Tree size vs. $1/\epsilon$ (log–log) with $n = 20$, $\delta = 0.1$, target depth 4 (16 leaves) for the balanced DT and 16 leaves for the unbalanced DT, 6 repetitions per configuration.

*(b)* Tree size vs. dimension $n \in \{3, \ldots, 7\}$ with $\epsilon = 0.15$, $\delta = 0.1$, target depth 3 (up to 8 leaves), 6 repetitions per configuration.

*Figure 2.* Empirical evaluation of the practical, parameter-free top-down heuristic (Algorithm 2) on a balanced DT (left panel of each subfigure) and an unbalanced DT (right panel of each subfigure). Curves correspond to product distributions with bias $p \in \{0.5, 0.3, 0.1\}$; error bars show $\pm 1$ standard deviation across the 6 repetitions.

# E. Worked Examples: Balanced and Path-Like Trees

In this appendix we work through two examples that illustrate the role of the maximum depth $D_{opt}$ and the average depth $\Delta_{opt}$ in the bound of Theorem 1.1, and in particular show why the mixed dependence on the two parameters can be exponentially tighter than a bound stated in $D_{opt}$ alone.

### E.1. Balanced (Full Binary) Tree

Suppose the optimal decision tree $T_{opt}$ for $f$ is a complete binary tree of size $s$ over $n = \log_2 s$ variables. Then every leaf is at depth $\log_2 s$, so $D_{opt} = \Delta_{opt} = \log_2 s$. Substituting into Theorem 1.1 gives

$$\text{size}(T) \ \leq \ \max\left(\left(\frac{e\,\Delta_{opt}}{\epsilon\,D_{opt}}\right)^{\Delta_{opt} D_{opt}}, \ e^{\Delta_{opt} D_{opt}}\right) \ = \ \left(\frac{e}{\epsilon}\right)^{\log^2 s} \ = \ s^{\log s \cdot \log(e/\epsilon)},$$

which is quasi-polynomial in $s$. This is the regime where our bound coincides asymptotically with — and slightly improves on — the bound of (Blanc et al., 2020).

### E.2. Path-Like (Highly Unbalanced) Tree

At the other extreme, consider a *path tree* (sometimes called a chain or comb tree) on $n$ variables under the uniform distribution over $\{0, 1\}^n$: the root queries $x_1$; one of its children is a leaf and the other queries $x_2$; one of *that* child's children is a leaf and the other queries $x_3$; and so on. Thus the path tree has $n + 1$ leaves, the deepest of which sits at depth $n$.

**Maximum vs. average depth.** The maximum depth is $D_{opt} = n$. To compute the average depth, observe that under the uniform distribution each internal node has a $1/2$ probability of routing the input into its leaf child; hence a random input reaches the leaf at depth $k$ with probability $2^{-k}$ for $k = 1, \ldots, n-1$, and reaches the deepest leaf with probability $2^{-(n-1)}$. Therefore

$$\Delta_{opt} \ = \ \sum_{k=1}^{n-1} k \cdot 2^{-k} \ + \ n \cdot 2^{-(n-1)} \ \leq \ \sum_{k=1}^{\infty} k \cdot 2^{-k} \ = \ 2.$$

So while $D_{opt} = n$ grows linearly with the number of variables, $\Delta_{opt}$ is bounded by an absolute constant.

**Why the mixed dependence helps.**    Plugging into Theorem 1.1 yields a size bound dominated by $\Delta_{opt} D_{opt} \leq 2n$ in the exponent. By contrast, a bound in terms of $D_{opt}^2 = n^2$ alone would be exponentially worse: the resulting size would be $\exp(\Theta(n^2))$ rather than $\exp(\Theta(n))$. This example illustrates why our analysis tracks both $D_{opt}$ and $\Delta_{opt}$ separately: in any regime where the leaves of $T_{opt}$ are not all equally deep, $\Delta_{opt}$ can be dramatically smaller than $D_{opt}$ and the mixed dependence is a strict improvement over a bound stated in $D_{opt}$ alone.

