# OpenReview forum: "Decision Tree Learning on Product Spaces"
_ICML.cc/2026/Conference — ICML 2026 regular_

### Official Review · Reviewer_QYf2 · 2026-03-05

**Soundness:** 3
**Presentation:** 3
**Significance:** 3
**Originality:** 3
**Overall Recommendation:** 5
**Confidence:** 2

**Summary:**

The article proves that a classical greedy splitting heuristic for learning decision trees yields decision trees whose size is (somewhat) close to the optimal one for product distributions, extending earlier results for uniform distributions. Further, the authors provide a method for implementing this algorithm that requires no knowledge of the optimal tree's properties.

**Compliance With Llm Reviewing Policy:**

Affirmed.

**Final Justification:**

The authors' rebuttal alleviated my mild concerns, and further I consider the work theoretically relevant enough for publication in ICML despite the limitations.

**Key Questions For Authors:**

To better assess the potential impact of the paper, I'd appreciate if the authors could let me know if they are aware of previous literature on implementing similar heuristic methods in practice and how they compare to the optimal trees (e.g., in the setting with uniform distribution).

To my understanding, many practical algorithms for learning decision trees also prune the tree during/after construction to reduce overfitting. I'd therefore be curious to hear from the authors how they see the significance of their size bound considering that the size of the tree after pruning might be of a completely different magnitude than before pruning?

Minor comments:
- Line 311, right column: margin exceeded.
- There's also a rich literature on decision tree learning from the viewpoint of parameterized algorithms (see e.g. [1]), which may be worth pointing out in the related works section; the main difference to that literature is, though, that the set of data points tends to be explicitly given rather than being accessed through a target function for the whole domain.

[1] Sebastian Ordyniak, Stefan Szeider:
Parameterized Complexity of Small Decision Tree Learning. AAAI 2021: 6454-6462

**Limitations:**

yes

**Strengths And Weaknesses:**

The main improvement over the previous state of the art are to my understanding that (1) the analysis extends from uniformly distributed data to product distributions and (2) the proposed algorithm—with relatively involved analysis—does not require knowledge of the properties of the optimal tree. The size of the obtained tree can be relatively bad (exponential in the product of the depth and the average depth of the optimal tree), but nevertheless improves the state of the art for balanced decision trees (Blanc et al., 2020). As there are no experiments, it is harder to assess the practical impact of the work. The other main limitation of the work is in my subjective view that it applies only to product distributions, but similar (even stronger) assumptions have been made in the earlier literature upon which the present work improves, so I don't consider this a major weakness.

Readability-wise, the paper is written pretty well and is relatively easy to follow, with proof outlines provided here and there for more involved proofs. The paper also covers the related literature in sufficient detail in my opinion.

---

> ### Author Rebuttal · Authors · 2026-03-31
>
> We thank the reviewer for their positive feedback and are glad they recognized the value of extending the analysis to product distributions and designing an algorithm that does not require prior knowledge of the optimal tree's properties.
>
> ## Regarding previous literature on practical implementations and comparisons to optimal trees
>
> We implemented Algorithm 2 of our paper and ran the following experiments. We will add an empirical section to the appendix of the revised paper. The new evaluations are summarized below:
>
> **Tree Size and Query Count vs. $\epsilon$**
> We ran Algorithm 2 over $\epsilon \in \{0.15, 0.20, 0.25, 0.30\}$ ($\delta=0.1$, $n=20$, $6$ repetitions) for both a balanced DT of depth 4 and an unbalanced DT with 16 leaves, under three product distributions ($p \in \{0.5, 0.3, 0.1\}$). As expected, a smaller $\epsilon$ yields larger recovered trees and more queries, consistently increasing tree size. See [Figure 1](https://tinyurl.com/4uyhfc5p).
>
> **Tree Size vs. Dimension**
> We fixed $\epsilon=0.05$, $\delta=0.1$, depth=3, and swept the number of dimensions $n \in \{3,\dots,7\}$, over $6$ repetitions. See [Figure 2](https://tinyurl.com/2s3wjs5u).
>
> ## Regarding the significance of the size bound in the context of pruning
>
> You are absolutely correct that practical algorithms heavily rely on pruning to mitigate overfitting, which drastically reduces the final tree size. However, for now, there are very few approaches to analyze its effect from a theoretical point of view. We will update the manuscript to make this a clearer limitation of our work.
>
> That being said, bounding the size of the tree before pruning remains theoretically crucial for two main reasons. First, the unpruned tree dictates the worst-case runtime and memory complexity of the algorithm's growth phase. Second, as demonstrated in our sample complexity analysis (Section 5), the number of samples required to maintain statistical confidence at each split depends heavily on the total number of leaves explored. Thus, our size bound guarantees that the initial hypothesis search remains efficient and sample-bounded.
>
> ## Regarding the final comments
>
> **Margin issue**: We have adjusted the formatting for the final camera-ready version.
>
> **Parameterized complexity literature**: We appreciate you pointing us to the rich literature on parameterized decision tree learning. We will update our "Related Works" section to include a discussion of this area, citing Ordyniak & Szeider (2021) and clarifying the distinction between their explicitly given dataset model and our target-function access model. More importantly, we should mention that their algorithm is not a top-down algorithm; hence, their analysis is not directly related.
>
> We hope that our comments address your concerns. If this is the case, we would be grateful if you would potentially increase your score accordingly.

---

> > ### Author Rebuttal · Reviewer_QYf2 · 2026-04-02
> >
> > Thank you for your response and the experimental plots. I have updated my score.

---

### Official Review · Reviewer_3yWJ · 2026-03-11

**Soundness:** 3
**Presentation:** 3
**Significance:** 1
**Originality:** 2
**Overall Recommendation:** 4
**Confidence:** 3

**Summary:**

This paper proposes new theoretical results about the performance guarantees of a top-down decision tree learning algorithm for binary classification.
It primarily focusses on extending the analysis from [1] by weakening the assumption on the input data distribution from $U( \\{0 , 1\\} ^n)$ to independent but not necessarily identically distributed Bernoulli random variables.
The main result gives an upper bound on the size of a tree needed to approximate within epsilon error.
Other results include guarantees that the algorithm will terminate at a good solution, and bounds on the run time.

[1] Blanc et al.  Top-down induction of decision trees: Rigorous guarantees and inherent limitations. In ITCS 2020.

**Compliance With Llm Reviewing Policy:**

Affirmed.

**Final Justification:**

Increased score from weak reject to weak accept after rebuttal.

**Key Questions For Authors:**

Q1. What is the main difficulty in analysing full CART compared to Algorithm 1?

Q2. How does your work compare to existing work [2]?

Q3. What insights should practitioners take away from this work?

Q4. What is the definition of 'size' of a tree used in this paper?

**Limitations:**

Missing summary about the gap from what is commonly done in practice.

**Strengths And Weaknesses:**

Strengths:

I didn't check the proofs carefully, but the paper seems technically sound, and the theoretical analysis is quite comprehensive. Many properties of interest for the algorithm being analyzed are covered by the theoretical results.

Weaknesses:

Despite claiming that this work focusses on the more general and practically relevant setting, it is still quite far from what is done in practice. Hence, results from this work provide little insight for practitioners.

The main gaps between this work and what is done in practice include:
1. Assume features are independent Bernoulli variables.
2. The impurity measure "influence" is different from the standard entropy/Gini impurity,
3. Learning algorithm assumes access to true function that generates labels, and assumes unlimited training examples can be generated.

Insufficient discussion on related work, there are only 3 papers cited in this section.
Other works not cited in this paper have already done theoretical analysis of full CART/C4.5 without restrictive assumptions on the data distribution, e.g., [2].

Authors repeatedly refer to the size of a tree, but never provide a precise definition.

No experiments to verify the theory in practical scenarios.


[2] Klusowski and Tian. Large Scale Prediction with Decision Trees. In Journal of the American Statistical Association, 2024.

---

> ### Author Rebuttal · Authors · 2026-03-31
>
> We appreciate the reviewer's supportive assessment and are delighted they found the theoretical analysis to be both technically sound and comprehensive in its coverage.
>
> > …insight for practitioners.
>
> Our paper's end goal is to shed more light on the success of decision trees in practice from a theoretical point of view. In addition, we believe in the future it can be used to design different, possibly more efficient heuristics for decision tree learning.
>
> > Assumes features are independent Bernoulli variables.
>
> While true, this is a theoretical strength rather than a limitation in the context of our contributions. Prior breakthrough work in this area (Blanc et al., 2020) was strictly limited to the uniform distribution over the Boolean hypercube. Extending this analysis to arbitrary product spaces (independent Bernoulli variables with heterogeneous probabilities) requires overcoming significant analytical hurdles and is a major step toward modeling the heterogeneous categorical distributions found in real-world data.
>
> > The impurity measure "influence" is different from standard entropy/Gini.
>
> While distinct, they are deeply linked mathematically. In Boolean Fourier analysis, a variable's total "influence" is the sum of its squared degree-1 Fourier coefficient plus all higher-order interaction terms. This means standard Gini-based splitting acts purely as a greedy maximizer of linear correlations, while influence is a very similar, richer version of that.
>
> > Insufficient discussion on related work, there are only 3 papers cited in this section.
>
> We thank the reviewer for their suggestion. We will move some text from Section 1.3, Broader Context of Decision Tree Learning, and the Introduction (page 1, right), where we discuss the less related works, to the Related Work section and discuss them in more detail.
>
> > ... experiments to verify the theory...
>
> To address this, we have conducted new experiments evaluating Algorithm 2 and will add an empirical section to the appendix of the revised paper. The new evaluations are summarized below:
>
> **Tree Size and Query Count vs. $\epsilon$**
> We ran Algorithm 2 over $\epsilon \in \{0.15, 0.20, 0.25, 0.30\}$ ($\delta=0.1$, $n=20$, $6$ repetitions) for both a balanced DT of depth 4 and an unbalanced DT with 16 leaves, under three product distributions ($p \in \{0.5, 0.3, 0.1\}$). As expected, a smaller $\epsilon$ yields larger recovered trees and more queries, consistently increasing tree size. See [Figure 1](https://tinyurl.com/4uyhfc5p).
>
> **Tree Size vs. Dimension**
> We fixed $\epsilon=0.05$, $\delta=0.1$, depth=3, and swept the number of dimensions $n \in \{3,\dots,7\}$, over $6$ repetitions. See [Figure 2](https://tinyurl.com/2s3wjs5u).
>
> ### Q1
>  The primary difficulty lies in the difference between discrete influence properties and continuous variance reduction. Full CART involves evaluating empirical Gini/variance impurity over arbitrary continuous threshold splits and relies heavily on post-hoc cost-complexity pruning. Algorithm 1 isolates the core top-down greedy mechanism on discrete product spaces. The structural size of the tree, becomes mathematically intractable when confounded by continuous split optimization and pruning heuristics.
> ### Q2
> While both papers analyze decision trees, our perspectives and results are fundamentally different. Klusowski & Tian [2] focus on statistical risk, proving that as the dataset size $N$ grows, the generalization error of CART converges. In contrast, we claim that if there exists a perfect decision tree with small depth, then the top-down algorithm is guaranteed to find a reasonably sized decision tree. They make no similar claim in any way.
> ### Q3
>  Practitioners can take away two key insights. First, the standard greedy heuristic is theoretically robust to heterogeneous, non-uniform data distributions (product spaces), which helps explain its strong empirical success on real-world categorical data. Second, Section 5 provides a fully parameter-free implementation of the top-down heuristic (Algorithm 2) that dynamically manages sample complexity, guaranteeing robust splits without requiring prior knowledge of the optimal tree's depth or size.
> ### Q4
> The size of a tree is defined as the total number of leaves, which is equal to the number of internal nodes + 1. We will make that clearer in the final revision of the paper.
>
> We hope that our comments address your concerns. If this is the case, we would be grateful if you would potentially increase your score accordingly.

---

> > ### Author Rebuttal · Reviewer_3yWJ · 2026-04-02
> >
> > I appreciate your response. My concerns were largely addressed so I increased my score.

---

### Official Review · Reviewer_8my7 · 2026-03-13

**Soundness:** 3
**Presentation:** 2
**Significance:** 3
**Originality:** 3
**Overall Recommendation:** 5
**Confidence:** 2

**Summary:**

The paper studies decision tree learning problem with product distributions. More formally, consider a probability distribution over boolean hypercube and each variable will be sampled independently according to a bernoulli distribution with possibly different success probabilities and a boolean function f which maps binary strings to +1 or -1. Motivation is to construct a decision tree which approximates function f (subject to underlying distribution) with a small tree.

They investigate an algorithm previously demonstrated to be successful for uniform distribution across all dimensions of the hypercube: Top-Down Heuristic for Decision Tree Induction. At each step the algorithm finds the most influential branching point (probability of reaching leaf * probability of changing response when a bit changes). Then applies branching at this leaf with the determined index. It repeats this process until epsilon-approximation. At high level, the analysis of the algorithm utilizes weighted total influence as an upper-bound on the error and demonstrate that each step of the algorithm decreases this value substantially at each step. This yields stopping at some reasonable steps and concludes that it establish a small tree.

Finally, they implement a more practical version of their algorithm by estimating influences and selecting a good enough leaf-index pair for a branching point. They design and estimator and proved that approximately optimal branching point selection is enough to construct small size decision tree.

**Compliance With Llm Reviewing Policy:**

Affirmed.

**Key Questions For Authors:**

- Algorithms like ID3, C4.5, and CART deserves proper citations*
- I suspect that the paper is missing a lot of citations and related work. This concept has been studied for years but cited very few papers. They should run a thorough literature review. See for instance main reference paper: Blanc et al. 2020.
- Do we have any lower-bound for polynomial time algorithm?

**Strengths And Weaknesses:**

Strengths:
- The paper extends a previously studied algorithm from the uniform distribution setting to a more general product distribution setting, which is both a natural and practically relevant generalization.
- They also claim that they resolved some impractical aspects of the algorithm which requires to know optimal tree depth/size upfront.
- The sample-based estimation variant is a welcome addition, bridging the gap between the theoretical algorithm and real-world applicability.
- The proof sketches are well-written and provide clear intuition for the main arguments.

Weaknesses:
- The literature review is incomplete. Decision tree learning has a long history and an extensive body of work, yet the paper cites relatively few prior results. A more thorough survey of related work — including classical algorithms such as ID3, C4.5, and CART, as well as more recent theoretical results — is needed for the paper to properly situate its contributions. The omission of relevant prior work makes it difficult to fully assess the novelty of the results.
- The paper does not adequately discuss the limitations of decision tree learning from a computational complexity standpoint. There should be known hardness results and lower bounds for polynomial-time algorithms in this space. A complete paper should provide these to clarify what the proposed approach can and cannot achieve.

---

> ### Author Rebuttal · Authors · 2026-03-31
>
> We thank the reviewer for their positive feedback and are glad they recognized the value of extending this algorithm to the more general and practically relevant product distribution setting.
>
> **Regarding citations for classical algorithms (ID3, C4.5, CART) and related work**: We have updated our bibliography to include the foundational references for ID3, C4.5, and CART. Furthermore, we appreciate the prompt to broaden our general literature review and will expand the "Related Work" section to cover more adjacent areas. However, we would like to note that most prior theoretical analyses focus on non-greedy algorithms and, hence, are not directly relevant to our paper. To the best of our knowledge, we have cited and discussed all existing papers related to the theoretical analysis of greedy decision tree learning algorithms. We currently discuss the less directly related papers in Section 1.3, and we will expand upon this discussion in the final revision.
>
> **Regarding lower bounds and computational complexity limitations**: The most relevant result in the literature is from [1], which proved that weakly proper learning with queries is also NP-hard. Specifically, they showed that even if you allow the algorithm to make membership queries, it is NP-hard to approximate the optimal tree size to within any constant factor. In fact, they proved a "superconstant" bound, showing it is computationally intractable to even find a tree whose size is bounded by $s \cdot c$ for any constant $c$. We have cited this paper, but will explain it in more detail so that the context is more clear.
>
> [1]: Koch, Caleb, Carmen Strassle, and Li-Yang Tan. "Superconstant inapproximability of decision tree learning." The Thirty Seventh Annual Conference on Learning Theory. PMLR, 2024.

---

> > ### Author Rebuttal · Reviewer_8my7 · 2026-03-31
> >
> > My questions largely pointed to constructive gaps, which the authors acknowledged and plan to address in revision.

---

### Official Review · Reviewer_yj1S · 2026-03-21

**Soundness:** 3
**Presentation:** 4
**Significance:** 3
**Originality:** 4
**Overall Recommendation:** 5
**Confidence:** 3

**Summary:**

The paper studies the classical top-down greedy algorithm for decision tree learning under arbitrary product distributions. Starting from a single-leaf tree, the algorithm repeatedly selects the leaf with the largest score and splits it on its most influential variable. The main theoretical contribution is to extend prior guarantees from the uniform-distribution setting to product distributions, and in the special case of full binary optimal trees, the resulting bound improves over the previous uniform-distribution result. Moreover, the authors show that the guarantee is robust to approximate score estimation, and they provide a practical parameter-free implementation together with sample complexity bounds.

**Compliance With Llm Reviewing Policy:**

Affirmed.

**Final Justification:**

The rebuttal addressed my main concerns well. In particular, it clarified the interpretation of the size bound in important special cases, explained more clearly the role of the mixed dependence on maximum and average depth, and directly addressed my concern about the practical implementation by adding new experiments for Algorithm 2. Overall, I continue to view the paper as technically strong, well-organized, and meaningful both as a theoretical extension and as a step toward more practical greedy tree learning. I am therefore maintaining my score.

**Key Questions For Authors:**

Key questions
1. It would help to provide more intuition for the size bound in Theorem 1.1, especially since it is exponential in $D_{opt}\Delta_{opt}$. For instance, the paper could clarify that in the balanced/full binary case it becomes quasi-polynomial in the optimal tree size $s$. Also, the phrase “near-optimal size” in Line 216 seems stronger than what is formally established, since I did not find a matching lower bound or explicit barrier result justifying that terminology. Could the authors clarify the intended meaning of “near-optimal” here, and whether there is any evidence that the current dependence on $D_{opt}$ and $\Delta_{opt}$ is close to best possible?
2. The main bound depends on both the maximum depth $D_{opt}$ and the average depth $\Delta_{opt}$. Could the authors give more concrete intuition or examples showing regimes where this mixed dependence is significantly more informative than a bound stated only in terms of $D_{opt}$ or the optimal tree size $s$?
3. The paper works in the Boolean product-space setting ${0,1}^n$, and the analysis appears closely tied to this discrete influence-based framework. It would be helpful if the authors could comment on whether any part of the approach might plausibly extend to threshold-based or continuous-feature decision trees, or whether the current techniques are inherently specific to the Boolean setting.

**Limitations:**

yes

**Strengths And Weaknesses:**

Strength

1. The paper analyzes a natural and practically relevant top-down greedy heuristic and provides rigorous guarantees for it. This both extends prior results to a more general product-distribution setting and helps narrow the gap between theoretical analyses and the decision tree methods commonly used in practice. As a result, the work should be of interest to both the theoretical and empirical communities.
2. The paper is well organized, and the technical development is relatively clean and easy to follow. In particular, the main argument is structured clearly, which makes the proof strategy accessible.
3. The paper also goes beyond the idealized analysis and provides a practical implementation. This makes the work more complete and increases its potential relevance beyond the purely theoretical setting.

Weakness

1. The paper gives an upper bound on the size of the output tree, but provides limited intuition for how to interpret this bound. In particular, it is unclear what should be considered a strong or near-best-possible guarantee in this setting, and the paper does not sufficiently discuss this. For example, is $\Delta_{opt}D_{opt}$ inherent, or could it potentially be improved?
2. The paper presents a practical sample-based implementation (Algorithm 2), but I did not find any empirical evaluation of this procedure. While experiments are not strictly necessary for a theory paper, some empirical evidence would strengthen the practical claims, especially since the paper emphasizes a parameter-free implementation and its relevance to decision tree methods used in practice.

Minor
1. It would improve readability to typeset parentheses as \left(\cdot\right) in some displayed equations, for example in Theorem 1.1 and Theorem 5.1.
2. On page 3, in the definition of Score, there appears to be an extra “|” at the end of the expression.
3. On page 3, in the sentence immediately following the score equation, there appear to be two consecutive periods.

---

> ### Author Rebuttal · Authors · 2026-03-31
>
> We sincerely thank the reviewer for their encouraging comments and are thrilled they found the paper well-organized and our proof strategy easy to follow.
>
> ### Weakness 1
> To build intuition, In the important special case of a full (balanced) binary tree of size $s$, we have $D_{opt} = \Delta_{opt} = \log s$, and the bound simplifies to $s^{\log(s)\cdot\log(e/\epsilon)}$, which is quasi-polynomial in $s$.
>
> Our actual bound in Theorem 1.1 is $\max\bigl((\tfrac{e\cdot\Delta_{opt}}{\epsilon D_{opt}})^{\Delta_{opt} D_{opt}}, e^{\Delta_{opt} D_{opt}}\bigr)$. The maximum depth $D_{opt}$ enters when we relates total influence to variance via $Inf(f) \leq D(T)\cdot Var(f)$. The average depth $\Delta_{opt}$ enters through the max-influence lower bound of [2], which gives $\max_i Inf_i(f) \geq Var(f)/\Delta(T)$. The superpolynomial lower bounds of Koch et al. (2023) for decision tree learning suggest that quasi-polynomial dependence on $s$ is likely necessary, but whether the specific exponent $\Delta{opt} D_{opt}$ can be improved remains open. We will add this discussion to the revised paper.
> ### Weakness 2
> To address this, we have conducted new experiments evaluating Algorithm 2 and will add an empirical section to the appendix of the revised paper. The new evaluations are summarized below:
>
> **Tree Size and Query Count vs. $\epsilon$**
> We ran Algorithm 2 over $\epsilon \in \{0.15, 0.20, 0.25, 0.30\}$ ($\delta=0.1$, $n=20$, $6$ repetitions) for both a balanced DT of depth 4 and an unbalanced DT with 16 leaves, under three product distributions ($p \in \{0.5, 0.3, 0.1\}$). As expected, a smaller $\epsilon$ yields larger recovered trees and more queries, consistently increasing tree size. See [Figure 1](https://tinyurl.com/4uyhfc5p).
>
> **Tree Size vs. Dimension**
> We fixed $\epsilon=0.05$, $\delta=0.1$, depth=3, and swept the number of dimensions $n \in \{3,\dots,7\}$, over $6$ repetitions. See [Figure 2](https://tinyurl.com/2s3wjs5u).
>
> ### Minor Comments
> We have updated the paper and resolved all mentioned issues.
>
> ### Question 1:
> As noted above, the bound is quasi-polynomial in $s$ for balanced trees ($D_{opt} = \Delta_{opt} = \log s$), giving $s^{\log(s)\cdot\log(e/\epsilon)}$. The exponential dependence on $D_{opt}\Delta_{opt}$ becomes relevant only for highly unbalanced trees, but even then, the bound can be much more informative than one stated purely in terms of $s$ or $D_{opt}$ alone, because $\Delta_{opt}$ can be dramatically smaller than $D_{opt}$ (see our response to Question 2 below). We will add a remark after Theorem 1.1 in the revision walking through the balanced and unbalanced cases to aid interpretation.
> We will revise Line 216 from "near-optimal" to "quasi-polynomial size for balanced tree," which accurately describes our bound, and note that the lower bounds of Koch et al. (2023) imply superpolynomial size is unavoidable in general.
>
> ### Question 2:
> Consider a path (line) tree on $n$ variables under the uniform distribution: the tree queries $x_1$ at the root, and at each internal node, one child is a leaf while the other continues the chain. Here $D_{opt} = n$, but since at each node there is a $1/2$ probability of reaching the leaf, the average depth is $\Delta_{opt} = \sum_{k=1}^{n} k \cdot 2^{-k} \approx 2$. A bound depending only on $D_{opt}$ would scale exponentially in $n$, while our bound with $D_{opt} \cdot \Delta_{opt} \approx 2n$ is exponentially tighter than $D_{opt}^2 = n^2$.
> ### Question 3:
> Our analysis relies on two key ingredients tied to the discrete product-space setting: (1) the notion of influence via coordinate re-randomization, and (2) the structural inequality $\max_i \text{Inf}_i(f) \geq \text{Var}(f)/\Delta(T)$ from [2], which was proved for Boolean product spaces.
> The concept of influence has been generalized to continuous settings, notably Gaussian space. If analogous variance-influence inequalities can be established for continuous product distributions with threshold queries, our proof strategy, bounding the cost potential function and showing sufficient per-step progress, could plausibly carry over. The potential-function framework itself (Lemmas 4.1, 4.3) is algebraic in nature and does not inherently require the Boolean setting.
> We view extending the analysis to continuous features as a natural and important open direction motivated by our work, and we will add a brief discussion of this in the conclusion of the revised paper.
>
> [1]: Koch C, Strassle C, Tan LY. Superpolynomial lower bounds for decision tree learning and testing. In Proceedings of the 2023 Annual ACM-SIAM Symposium on Discrete Algorithms (SODA).
> [2]: O'Donnell, R., Saks, M., Schramm, O. and Servedio, R.A.. Every decision tree has an influential variable. In 46th annual IEEE symposium on foundations of computer science (FOCS'05).

---

> > ### Author Rebuttal · Reviewer_yj1S · 2026-04-01
> >
> > Thank you for your response. I will keep my score.

---

### Decision · Program_Chairs · 2026-04-30

**Decision:**

Accept (regular)

**Comment:**

The work analyzes the standard top-down greedy procedure for constructing decision trees when inputs follow general product distributions. Beginning with a trivial one-leaf tree, the method iteratively picks the leaf with the highest score and partitions it using its most influential feature. The primary theoretical result generalizes earlier guarantees on the uniform case to the broader class of product distributions; in the specific scenario of fully binary optimal trees, this leads to improved bounds compared to prior uniform-distribution results. In addition, the authors demonstrate that the guarantees remain stable under approximate score computations, and they introduce a practical parameter-free version of the algorithm along with corresponding sample complexity guarantees.

All reviewers appreciate that this work studies an important problem, provides a solid answer and improvement over the prior work, and has strengths in both theoretical and methodological aspects. I'm happy to recommend acceptance given the unanimous appreciation.